# Class-Conditional Conformal Prediction
# with Many Classes

**Tiffany Ding**
University of California, Berkeley
tiffany_ding@berkeley.edu

**Anastasios N. Angelopoulos**
University of California, Berkeley
angelopoulos@berkeley.edu

**Stephen Bates**
MIT
s_bates@mit.edu

**Michael I. Jordan**
University of California, Berkeley
jordan@cs.berkeley.edu

**Ryan J. Tibshirani**
University of California, Berkeley
ryantibs@berkeley.edu

## Abstract

Standard conformal prediction methods provide a marginal coverage guarantee, which means that for a random test point, the conformal prediction set contains the true label with a user-specified probability. In many classification problems, we would like to obtain a stronger guarantee—that for test points *of a specific class*, the prediction set contains the true label with the same user-chosen probability. For the latter goal, existing conformal prediction methods do not work well when there is a limited amount of labeled data per class, as is often the case in real applications where the number of classes is large. We propose a method called *clustered conformal prediction* that clusters together classes having "similar" conformal scores and performs conformal prediction at the cluster level. Based on empirical evaluation across four image data sets with many (up to 1000) classes, we find that clustered conformal typically outperforms existing methods in terms of class-conditional coverage and set size metrics.

## 1 Introduction

Consider a situation in which a doctor relies on a machine learning system that has been trained to output a set of likely medical diagnoses based on CT images of the head. Suppose that the system performs well on average and is able to produce prediction sets that contain the actual diagnosis with at least 0.9 probability. Upon closer examination, however, the doctor discovers the algorithm only predicts sets containing common and relatively benign conditions, such as {normal, concussion}, and the sets never include less common but potentially fatal diseases, such as intracranial hemorrhage. In this case, despite its high marginal (average) performance, the doctor would not want to use such an algorithm because it may lead to patients missing out on receiving critical care. The core problem is that even though the average performance of the algorithm is good, the performance for some classes is quite poor.

Conformal prediction (Vovk et al., 2005) is a method for producing set-valued predictions that serves as a wrapper around existing prediction systems, such as neural networks. Standard conformal prediction takes a black-box prediction model, a calibration data set, and a new test example $X_{\text{test}} \in \mathcal{X}$ with unknown label $Y_{\text{test}} \in \mathcal{Y}$ and creates a prediction set $\mathcal{C}(X_{\text{test}}) \subseteq \mathcal{Y}$ that satisfies *marginal coverage*:

$$\mathbb{P}(Y_{\text{test}} \in \mathcal{C}(X_{\text{test}})) \geq 1 - \alpha, \tag{1}$$

for a coverage level $\alpha \in [0, 1]$ chosen by the user. However, as the example above shows, the utility of these prediction sets can be limited in some real applications. In classification, which we study in

37th Conference on Neural Information Processing Systems (NeurIPS 2023).

this paper, the label space $\mathcal{Y}$ is discrete, and it is often desirable to have *class-conditional coverage*:

$$\mathbb{P}(Y_{\text{test}} \in \mathcal{C}(X_{\text{test}}) \mid Y_{\text{test}} = y) \geq 1 - \alpha, \quad \text{for all } y \in \mathcal{Y}, \tag{2}$$

meaning that every class $y$ has at least $1 - \alpha$ probability of being included in the prediction set when it is the true label. Note that (2) implies (1). Predictions sets that only satisfy (1) may neglect the coverage of some classes, whereas the predictions sets in (2) are effectively "fair" with respect to all classes, even the less common ones.

Standard conformal prediction, which we will refer to as STANDARD, does not generally provide class-conditional coverage. We present a brief case study to illustrate.

**ImageNet case study.** Running STANDARD using a nominal coverage level of 90% on 50,000 examples sampled randomly from ImageNet (Russakovsky et al., 2015), a large-scale image data set described later in Section 3, yields prediction sets that achieve very close to the correct marginal coverage (89.8%). However, this marginal coverage is achieved by substantially undercovering some classes and overcovering others. For example, water jug is severely undercovered: the prediction sets only include it in 50.8% of the cases where it is the true label. On the other hand, flamingo is substantially overcovered: it achieves a class-conditional coverage of 99.2%. This underscores the need for more refined methods in order to achieve the class-conditional coverage defined in (2).

In principle, it is possible to achieve (2) by splitting the calibration data by class and running conformal prediction once for each class (Vovk, 2012). We refer to this as CLASSWISE. However, this procedure fails to be useful in many real applications since data-splitting can result in very few calibration examples for each class-level conformal prediction procedure. This typically happens in problem settings where we have many classes but limited data; in such settings, the classwise procedure tends to be overly conservative and produces prediction sets that are too large to be practically useful. We note that previous papers (Shi et al., 2013; Löfström et al., 2015) on class-conditional conformal have not focused on the many classes regime and have instead studied binary or at most 10-way classification tasks.

In this work, we focus on the challenging limited-data, many-class classification setting and we develop a method targeted at class-conditional coverage. Our method mitigates the problems that arise from data-splitting by clustering together classes that have similar score distributions and combining the calibration data for those classes. Figure 1 illustrates how the method we propose strikes a balance between STANDARD and CLASSWISE. As we will later show in our experiments, this can improve class-conditional coverage in many situations.

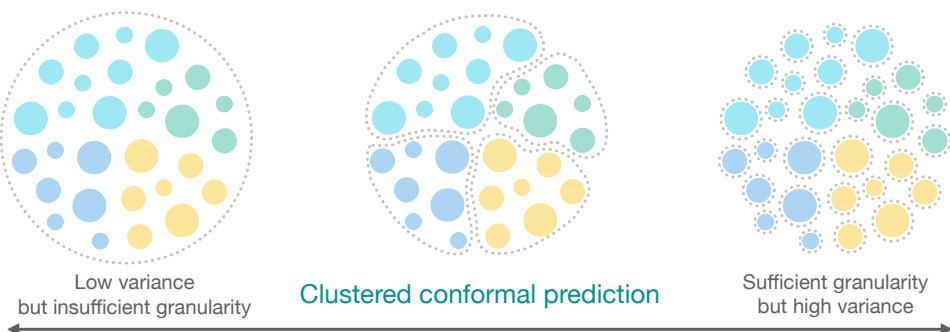

Figure 1: A schematic comparison of conformal prediction methods, including the CLUSTERED method we propose. Each colored circle represents the calibration data for a particular class. Existing methods fall on extremes of the spectrum: STANDARD is very stable because it groups data for all classes together, but it is not able to treat classes differently when needed; CLASSWISE has high granularity, but it splits all of the data by class and consequently has high variance in limited-data settings; CLUSTERED strikes a balance by grouping together data for "similar" classes.

**Our contributions.** We make three advances in tackling the problem of class-conditional coverage.

- We propose an extension of conformal prediction called *clustered conformal prediction* that often outperforms standard and classwise conformal, in terms of class-conditional coverage, when there is limited calibration data available per class.

- We present a comprehensive empirical evaluation of class-conditional coverage for conformal methods on four large-scale classification data sets, each with many (100 to 1000) classes.

- We provide general guidelines to practitioners for how to choose an appropriate conformal prediction method in order to achieve class-conditional coverage in the problem setting at hand.

## 1.1 Related work

**Mondrian conformal prediction.** We work in the split-conformal prediction framework (Papadopoulos et al., 2002; Lei et al., 2018) in which the training data set (to train the base classifier) and calibration data set are disjoint. Mondrian conformal prediction (MCP) is a general procedure that encompasses many kinds of conditional conformal prediction (Vovk et al., 2005). For any chosen grouping function $G : \mathcal{X} \times \mathcal{Y} \to \mathcal{G}$ where $\mathcal{G}$ denotes the set of all groups, MCP provides coverage guarantees of the form $\mathbb{P}(Y_{\text{test}} \in \mathcal{C}(X_{\text{test}}) \mid G(X_{\text{test}}, Y_{\text{test}}) = g) \geq 1 - \alpha$ for all groups $g \in \mathcal{G}$. The high-level idea behind MCP is to split the calibration data by group and then run conformal prediction on each group. The CLASSWISE conformal procedure is a special case of MCP where each value of $\mathcal{Y}$ defines a group (this is also sometimes referred to as label-conditional conformal prediction).

To the best of our knowledge, previous work has not focused on class-conditional coverage of conformal methods in the many-classes, limited-data regime that is common to many real classification applications. Löfström et al. (2015) present an empirical study of STANDARD and CLASSWISE on binary, 3-way, and 4-way classification data sets (with a focus on the binary setting). For class-imbalanced problems, they find that STANDARD tends to overcover the majority class, and undercover the minority class. Shi et al. (2013) consider CLASSWISE on the MNIST and USPS data sets, which are 10-way classification problems. Sun et al. (2017) use a cross-fold variant of CLASSWISE for binary classification on imbalanced bioactivity data. Hechtlinger et al. (2018) run class-conditional experiments on the 3-class Iris data set, and provide preliminary ideas on how to incorporate interactions between classes using density estimation. Guan and Tibshirani (2022) use CLASSWISE in an outlier detection context. Sadinle et al. (2019) consider CLASSWISE with a modification to avoid empty prediction sets and perform experiments on 3-way or 10-way classification tasks. To reiterate, the aforementioned papers all focus on the regime that is generally favorable to CLASSWISE, where data is abundant relative to the number of classes. Our work focuses on the limited-data regime that often arises in practice.

**Other types of conditional conformal prediction.** "Conditional coverage" is a term that, within the conformal prediction literature, often refers to coverage at a particular input value $X = x$. It is known to be impossible to achieve coverage conditional on $X = x$, for all $x$, without invoking further distributional assumptions (Lei and Wasserman, 2014; Vovk, 2012; Barber et al., 2021). However, approximate $X$-conditional coverage can be achieved in practice by designing better score functions (Romano et al., 2019, 2020b; Angelopoulos et al., 2022) or by modifying the conformal procedure itself (Romano et al., 2020a; Guan, 2023; Gibbs et al., 2023). Our work draws inspiration from the latter camp, but our aim is to achieve class-conditional ($Y$-conditional) coverage rather than $X$-conditional coverage. Note that $X$-conditional coverage is hard to interpret in settings in which $Y$ is not random given $X = x$ (e.g., in image classification, if $x$ is an image of a dog, then the true label is always $y = $ dog, with no intrinsic randomness after conditioning on $X = x$). In such settings, it is more natural to consider class-conditional coverage.

## 1.2 Preliminaries

We work in the classification setting in which each input $X_i \in \mathcal{X}$ has a class label $Y_i \in \mathcal{Y}$, for some discrete set $\mathcal{Y}$. Let $\{(X_i, Y_i)\}_{i=1}^N$ denote a *calibration data set*, where each $(X_i, Y_i) \overset{\text{iid}}{\sim} F$. Given a new independent test point $(X_{\text{test}}, Y_{\text{test}}) \sim F$, our goal is to construct (without knowledge of $Y_{\text{test}}$) a prediction set $\mathcal{C}(X_{\text{test}})$ that satisfies (2) for some user-specified $\alpha \in [0, 1]$.

Let $s : \mathcal{X} \times \mathcal{Y} \to \mathbb{R}$ denote a *(conformal) score function*, where we take $s(x, y)$ to be negatively oriented, which means that lower scores indicate a better agreement between the input $x$ and the proposed class label $y$. The score function is typically derived from a pre-trained classifier $f$ (a simple example to keep in mind is $s(x, y) = 1 - f_y(x)$, where $f_y(x)$ represents the $y^{\text{th}}$ entry of the softmax vector output by $f$ for the input $x$). For brevity, we denote the score of the $i^{\text{th}}$ calibration data point as $s_i = s(X_i, Y_i)$. For $\tau \in [0, 1]$ and a finite set $A \subseteq \mathbb{R}$, let $\text{Quantile}(\tau, A)$ denote the smallest $a \in A$ such that $\tau$ fraction of elements in $A$ are less than or equal to $a$. For $\tau > 1$, we take $\text{Quantile}(\tau, A) = \infty$.

With this notation, the STANDARD conformal prediction sets are given by

$$\mathcal{C}_{\text{STANDARD}}(X_{\text{test}}) = \{y : s(X_{\text{test}}, y) \leq \hat{q}\},$$

where

$$\hat{q} = \text{Quantile}\left(\frac{\lceil (N+1)(1-\alpha) \rceil}{N}, \{s_i\}_{i=1}^N\right).$$

These prediction sets are guaranteed to satisfy (1) (Vovk et al., 2005). We can interpret $\hat{q}$ as a finite-sample adjusted $(1-\alpha)$-quantile of the scores in the calibration data set. Note that all $N$ data points are used for computing a single number, $\hat{q}$.

In contrast, the CLASSWISE procedure computes a separate quantile for every class. Let $\mathcal{I}^y = \{i \in [N] : Y_i = y\}$ be the indices of examples in the calibration data set that have label $y$. The CLASSWISE conformal prediction sets are given by

$$\mathcal{C}_{\text{CLASSWISE}}(X_{\text{test}}) = \{y : s(X_{\text{test}}, y) \leq \hat{q}^y\},$$

where

$$\hat{q}^y = \text{Quantile}\left(\frac{\lceil (|\mathcal{I}^y|+1)(1-\alpha) \rceil}{|\mathcal{I}^y|}, \{s_i\}_{i \in \mathcal{I}^y}\right).$$

These prediction sets are guaranteed to satisfy (2) (Vovk, 2012). However, the quantile $\hat{q}^y$ is only computed using a subset of the data of size $|\mathcal{I}_y|$, which might be quite small. Importantly, for any class $y$ for which $|\mathcal{I}^y| < (1/\alpha) - 1$, we will have $\hat{q}^y = \infty$, hence any prediction set generated by CLASSWISE will include $y$, no matter the values of the conformal scores.

The advantage of STANDARD is that we do not need to split up the calibration data to estimate $\hat{q}$, so the estimated quantile has little noise even in limited-data settings; however, it does not in general yield class-conditional coverage. Conversely, CLASSWISE is guaranteed to achieve class-conditional coverage, since we estimate a different threshold $\hat{q}^y$ for every class; however, these estimated quantiles can be very noisy due to the limited data, leading to erratic behavior, such as large sets.

When we assume the scores are almost surely distinct, the exact distribution of the CLASSWISE class-conditional coverage of class $y$ given a fixed calibration set is (Vovk, 2012; Angelopoulos and Bates, 2023):

$$\mathbb{P}\big(Y_{\text{test}} \in \mathcal{C}_{\text{CLASSWISE}}(X_{\text{test}}) \mid Y_{\text{test}} = y, \{(X_i, Y_i)\}_{i=1}^N\big) \sim \text{Beta}(k_\alpha^y, |\mathcal{I}^y| + 1 - k_\alpha^y),$$

where $k_\alpha^y = \lceil (|\mathcal{I}^y|+1)(1-\alpha) \rceil$. Note that the probability on the left-hand side above is taken with respect to the test point $X_{\text{test}}$ as the only source of randomness, as we have conditioned on the calibration set. The beta distribution $\text{Beta}(k_\alpha^y, |\mathcal{I}^y| + 1 - k_\alpha^y)$ has mean $k_\alpha^y / (|\mathcal{I}^y| + 1)$ (which, as expected, is always at least $1 - \alpha$) and variance

$$\frac{k_\alpha^y(|\mathcal{I}^y| + 1 - k_\alpha^y)}{(|\mathcal{I}^y| + 1)^2(|\mathcal{I}^y| + 2)} \approx \frac{\alpha(1-\alpha)}{|\mathcal{I}^y| + 2},$$

which can be large when $|\mathcal{I}^y|$ is small. For example, if class $y$ only has 10 calibration examples and we seek 90% coverage, then the class-conditional coverage given a fixed calibration set is distributed as $\text{Beta}(10, 1)$, so there is probability $\approx 0.107$ that the coverage of class $y$ will be less than 80%. Somewhat paradoxically, the variance of the class-conditional coverage means that on a given realization of the calibration set, the CLASSWISE method can exhibit poor coverage on a substantial fraction of classes if the number of calibration data points per class is limited.

# 2 Clustered conformal prediction

With the goal of achieving the class-conditional coverage guarantee in (2), we introduce *clustered conformal prediction*. Our method strikes a balance between the granularity of CLASSWISE and the data-pooling of STANDARD by grouping together classes according to a clustering function. For each cluster, we calculate a single quantile based on all data in that cluster. We design the clustering algorithm so that clustered classes have similar score distributions, and therefore, the resulting cluster-level quantile is applicable to all classes in the cluster. Next, in Section 2.1, we formally describe the clustered conformal prediction method; then, in Section 2.2, we describe the clustering step in detail.

## 2.1 Meta-algorithm

To begin, we randomly split the calibration data set into two parts: the *clustering data set* $D_1 = \{(X_i, Y_i) : i \in \mathcal{I}_1\}$, for performing clustering, and a *proper calibration data set* $D_2 = \{(X_i, Y_i) : i \in \mathcal{I}_2\}$, for computing the conformal quantiles, where $|\mathcal{I}_1| = \lfloor \gamma N \rfloor$ and $|\mathcal{I}_2| = N - |\mathcal{I}_1|$ for some tuning parameter $\gamma \in [0, 1]$. Then, we apply a clustering algorithm to $D_1$ to obtain a *clustering function* $\hat{h} : \mathcal{Y} \to \{1, \ldots, M\} \cup \{\mathsf{null}\}$ that maps each class $y \in \mathcal{Y}$ to one of $M$ clusters or the "null" cluster (denoted $\mathsf{null}$). The reason that we include the null cluster is to handle rare classes that do not have sufficient data to be confidently clustered into any of the $M$ clusters. Details on how to create $\hat{h}$ are given in the next subsection.

After assigning classes to clusters, we perform the usual conformal calibration procedure within each cluster. Denote by $\mathcal{I}_2^y = \{i \in \mathcal{I}_2 : Y_i = y\}$ the indices of examples in $D_2$ with label $y$, and by $\mathcal{I}_2(m) = \cup_{y:\hat{h}(y)=m} \mathcal{I}_2^y$ the indices of examples in $D_2$ with labels in cluster $m$. The CLUSTERED conformal prediction sets are given by

$$\mathcal{C}_{\mathrm{CLUSTERED}}(X_{\mathrm{test}}) = \{y : s(X_{\mathrm{test}}, y) \leq \hat{q}(\hat{h}(y))\},$$

where, for $m = 1, \ldots, M$,

$$\hat{q}(m) = \mathrm{Quantile}\left( \frac{\lceil (|\mathcal{I}_2(m)| + 1)(1 - \alpha) \rceil}{|\mathcal{I}_2(m)|}, \{s_i\}_{i \in \mathcal{I}_2(m)} \right),$$

and

$$\hat{q}(\mathsf{null}) = \mathrm{Quantile}\left( \frac{\lceil (|\mathcal{I}_2| + 1)(1 - \alpha) \rceil}{|\mathcal{I}_2|}, \{s_i\}_{i \in \mathcal{I}_2} \right).$$

In words, for each cluster $m$, we group together examples for all classes in that cluster and then estimate a cluster-level quantile $\hat{q}(m)$. When constructing prediction sets, we include class $y$ in the set if the conformal score for class $y$ is less than or equal to the quantile for the cluster that contains $y$. For classes assigned to the null cluster, we compare the conformal score against the quantile that would be obtained from running STANDARD on the proper calibration set.

We now consider the properties of the CLUSTERED prediction sets. For all classes that do not belong to the null cluster, we have the following guarantee (the proof of this result, and all other proofs, are deferred to Appendix A).

**Proposition 1.** *The prediction sets $\mathcal{C} = \mathcal{C}_{\mathrm{CLUSTERED}}$ from CLUSTERED achieve cluster-conditional coverage:*

$$\mathbb{P}(Y_{\mathrm{test}} \in \mathcal{C}(X_{\mathrm{test}}) \mid \hat{h}(Y_{\mathrm{test}}) = m) \geq 1 - \alpha, \quad \textit{for all clusters } m = 1, \ldots, M.$$

This coverage result comes from the exchangeability between a test point *drawn from cluster $m$* and all of the calibration points that belong to cluster $m$. We get this exchangeability for free from the assumption that the calibration point and test points are sampled i.i.d. from the same distribution. Cluster-conditional coverage is a stronger guarantee than marginal coverage, but it is still not as strong as the class-conditional coverage property that we hope to achieve. However, cluster-conditional coverage implies class-conditional in an idealized setting: suppose we have access to an "oracle" clustering function $h^*$ that produces $M$ clusters such that, for each cluster $m = 1, \ldots, M$, the scores for all classes in this cluster are exchangeable (this would hold, for example, if all classes in the same cluster have the same score distribution). In this case, we have a guarantee on class-conditional coverage.

**Proposition 2.** *If $\hat{h} = h^*$, the "oracle" clustering function as described above, then the prediction sets from* CLUSTERED *satisfy class-conditional coverage* (2) *for all classes $y$ such that $h^*(y) \neq$ null.*

This coverage result arises because the oracle clustering function ensures exchangeability between a test point *drawn from class* $y$ and all of the calibration points that belong to the cluster to which $y$ belongs. To try to achieve this exchangeability, we need to construct the clusters carefully. Specifically, we want to design $\hat{h}$ so that the scores within each cluster $m = 1, \ldots, M$ are as close to identically distributed as possible, an idea we pursue next.

## 2.2 Quantile-based clustering

We seek to cluster together classes that have similar score distributions. To do so, we first summarize the empirical score distribution for each class via a vector of score quantiles evaluated at a discrete set of levels $\tau \in \mathcal{T} \subseteq [0, 1]$. In this embedding space, a larger distance between classes means their score distributions are more different. While there are various options for defining such an embedding, recall that to achieve class-conditional coverage, we want to accurately estimate the (finite-sample adjusted) $(1 - \alpha)$-quantile for the score distribution of each class. Thus, we want to group together classes with similar quantiles, which is what our embedding is designed to facilitate. After obtaining these embeddings, we can then simply apply any clustering algorithm, such as $k$-means.

In more detail, denote by $\mathcal{I}_1^y = \{i \in \mathcal{I}_1 : Y_i = y\}$ the indices of examples in $D_1$ with label $y$. We compute quantiles of the scores $\{s_i\}_{i \in \mathcal{I}_1^y}$ from class $y$ at the levels

$$\mathcal{T} = \left\{ \frac{\lceil (|\mathcal{I}^y| + 1)\tau \rceil}{|\mathcal{I}^y|} : \tau \in \{0.5, 0.6, 0.7, 0.8, 0.9\} \cup \{1 - \alpha\} \right\}.$$

and collect them into an embedding vector $z^y \in \mathbb{R}^{|\mathcal{T}|}$. If $|\mathcal{I}_1^y| < n_\alpha$, where $n_\alpha = (1/\min\{\alpha, 0.1\}) - 1$, then the uppermost quantile in $z^y$ will not be finite, so we simply assign $y$ to the null cluster. For a pre-chosen number of clusters $M$, we run $k$-means clustering with $k = M$ on the data $\{z^y\}_{y \in \mathcal{Y} \setminus \mathcal{Y}_{\text{null}}}$, where $\mathcal{Y}_{\text{null}}$ denotes the set of labels assigned to the null cluster. More specifically, we use a weighted version of $k$-means where the weight for class $y$ is set to $|\mathcal{I}_1^y|^{1/2}$; this allows the class embeddings computed from more data to have more influence on the cluster centroids. We denote the cluster mapping that results from this procedure by $\hat{h} : \mathcal{Y} \rightarrow \{1, \ldots, M\} \cup \{\text{null}\}$.

Of course, we cannot generally recover an oracle clustering function $h^*$ with the above (or any practical) procedure, so Proposition 2 does not apply. However, if the score distributions for the classes that $\hat{h}$ assigns to the same cluster are similar enough, then we can provide an approximate class-conditional coverage guarantee. We measure similarity in terms of the Kolmogorov-Smirnov (KS) distance, which is defined between random variables $X$ and $Y$ as $D_{\text{KS}}(X, Y) = \sup_{\lambda \in \mathbb{R}} |\mathbb{P}(X \leq \lambda) - \mathbb{P}(Y \leq \lambda)|$.

**Proposition 3.** *Let $S^y$ denote a random variable sampled from the score distribution for class $y$, and assume that the clustering map $\hat{h}$ satisfies*

$$D_{\text{KS}}(S^y, S^{y'}) \leq \epsilon, \quad \text{for all } y, y' \text{ such that } \hat{h}(y) = \hat{h}(y') \neq \text{null}.$$

*Then, for $\mathcal{C} = \mathcal{C}_{\text{CLUSTERED}}$ and for all classes $y$ such that such that $\hat{h}(y) \neq$ null,*

$$\mathbb{P}(Y_{\text{test}} \in \mathcal{C}(X_{\text{test}}) \mid Y_{\text{test}} = y) \geq 1 - \alpha - \epsilon.$$

To summarize, the two main tuning parameters used in the proposed method are $\gamma \in [0, 1]$, the fraction of points to use for the clustering step, and $M \geq 1$, the number of clusters to use in $k$-means. While there are no universal fixed values of these parameters that serve all applications equally well, we find that simple heuristics for setting these parameters often work well in practice, which we describe in the next section.

## 3 Experiments

We evaluate the class-conditional coverage of STANDARD, CLASSWISE, and CLUSTERED conformal prediction on four large-scale image data sets using three conformal score functions on each. Code for reproducing our experiments is available at `https://github.com/tiffanyding/class-conditional-conformal`.

## 3.1 Experimental setup

We run experiments on the ImageNet (Russakovsky et al., 2015), CIFAR-100 (Krizhevsky, 2009), Places365 (Zhou et al., 2018), and iNaturalist (Van Horn et al., 2018) image classification data sets, whose characteristics are summarized in Table 1. The first three have roughly balanced classes, the fourth, iNaturalist, has high class imbalance. We consider three conformal score functions: softmax, one minus the softmax output of the base classifier; APS, a score designed to improve $X$-conditional coverage; and RAPS, a regularized version of APS that often produces smaller sets. Precise definitions of the score functions are provided in Appendix B.1; we refer also to Romano et al. (2020b); Angelopoulos et al. (2021) for the details and motivation behind APS and RAPS. Throughout, we set $\alpha = 0.1$ for a desired coverage level of 90%.

Table 1: Description of data sets. The class balance metric is described precisely in Appendix B.3.

| Data set | ImageNet | CIFAR-100 | Places365 | iNaturalist |
|---|---|---|---|---|
| Number of classes | 1000 | 100 | 365 | 663* |
| Class balance | 0.79 | 0.90 | 0.77 | 0.12 |
| Example classes | mitten | orchid | beach | salamander |
| | triceratops | forest | sushi bar | legume |
| | guacamole | bicycle | catacomb | common fern |

*The number of classes in the iNaturalist data set can be adjusted by selecting which taxonomy level (e.g., species, genus, family) to use as the class labels. We use the species family as our label and then filter out any classes with $< 250$ examples in order to have sufficient examples to properly perform evaluation.

Our experiments all follow a common template. First, we fine-tune a pre-trained neural network as our base classifier (for details on model architectures, see Appendix B.2) on a small subset $D_{\text{fine}}$ of the original data, leaving the rest for calibration and validation purposes. We construct calibration sets of varying size by changing the average number of points in each class, denoted $n_{\text{avg}}$. For each $n_{\text{avg}} \in \{10, 20, 30, 40, 50, 75, 100, 150\}$, we construct a calibration set $D_{\text{cal}}$ by sampling $n_{\text{avg}} \times |\mathcal{Y}|$ examples without replacement from the remaining data $D_{\text{fine}}^c$ (where $^c$ denotes the set complement). We estimate the conformal quantiles for STANDARD, CLASSWISE, and CLUSTERED on $D_{\text{cal}}$. The remaining data $(D_{\text{fine}} \cup D_{\text{cal}})^c$ is used as the validation set for computing coverage and set size metrics. Finally, this process—splitting $D_{\text{fine}}^c$ into random calibration and validation sets—is repeated ten times, and the reported metrics are averaged over these repetitions.

**Details about clustering.** For CLUSTERED, we choose $\gamma \in [0, 1]$ (the fraction of calibration data points used for clustering) and $M \geq 1$ (the number of clusters) in the following way. First, we define $n_{\text{min}} = \min_{y \in \mathcal{Y}} |\mathcal{I}^y|$, the number of examples in the rarest class in the calibration set, and $n_\alpha = (1/\alpha) - 1$, the minimum sample size needed so that the finite-sample adjusted $(1 - \alpha)$-quantile used in conformal prediction is finite (e.g., $n_\alpha = 9$ when $\alpha = 0.1$). Now define $\tilde{n} = \max(n_{\text{min}}, n_\alpha)$ and let $K$ be the number of classes with at least $\tilde{n}$ examples. We then set $\gamma = K/(75 + K)$ and $M = \lfloor \gamma \tilde{n}/2 \rfloor$. These choices are motivated by two goals: we want $M$ and $\gamma$ to grow together (to find more clusters, we need more samples for clustering), and we want the proper calibration set to have at least 150 points per cluster on average; see Appendix B.4 for details on how the latter is achieved. Clustering is carried out by running $k$-means on the quantile-based embeddings, as described in Section 2.2; we use the implementation in `sklearn.cluster.KMeans`, with the default settings (Pedregosa et al., 2011).

## 3.2 Evaluation metrics

Denote the validation data set by $\{(X_i', Y_i')\}_{i=1}^{N'}$ (recall this is separate from the fine-tuning and calibration data sets) and let $\mathcal{J}^y = \{i \in [N'] : Y_i' = y\}$ be the indices of validation examples with label $y$. For any given conformal method, we define $\hat{c}_y = \frac{1}{|\mathcal{J}^y|} \sum_{i \in \mathcal{J}^y} \mathbb{1}\{Y_i' \in \mathcal{C}(X_i')\}$ as the empirical class-conditional coverage of class $y$. Our main evaluation metric is the *average class coverage gap* (CovGap):

$$\text{CovGap} = 100 \times \frac{1}{|\mathcal{Y}|} \sum_{y \in \mathcal{Y}} |\hat{c}_y - (1 - \alpha)|.$$

This measures how far the class-conditional coverage is from the desired coverage level of $1 - \alpha$, in terms of the $\ell_1$ distance across all classes (multiplied by 100 to put it on the percentage scale). We also measure the sharpness of the prediction sets by *average set size* (AvgSize):

$$\text{AvgSize} = \frac{1}{N'} \sum_{i=1}^{N'} |\mathcal{C}(X_i')|.$$

For a given class-conditional coverage level (a given CovGap), we want a smaller average set size.

### 3.3 Results

To begin, we investigate the CovGap of the methods on each data set, and display the results in Figure 2. In brief, CLUSTERED achieves the best or comparable performance across all settings. Restricting our attention to the baseline methods, note that the CovGap of STANDARD does not change much as we vary $n_{\text{avg}}$; this is as expected, because $n_{\text{avg}} \times |\mathcal{Y}|$ samples are being used to estimate the conformal quantile $\hat{q}$, which will be stable regardless of $n_{\text{avg}}$, provided $|\mathcal{Y}|$ is large. Conversely, the CovGap of CLASSWISE decreases significantly as $n_{\text{avg}}$ increases, because the classwise conformal quantiles $\hat{q}^y$ are volatile for small $n_{\text{avg}}$.

For the softmax score (left column of the figure), we see that CLUSTERED clearly outperforms STANDARD, and the gap widens with $n_{\text{avg}}$; further, CLUSTERED outperforms CLASSWISE for small enough values of $n_{\text{avg}}$ (that is, $n_{\text{avg}} < 75$ for ImageNet and Places365, $n_{\text{avg}} < 150$ for CIFAR-100, and $n_{\text{avg}} < 50$ for iNaturalist). The comparison between CLUSTERED and CLASSWISE is qualitatively similar for APS score (right column of the figure), with the former outperforming the

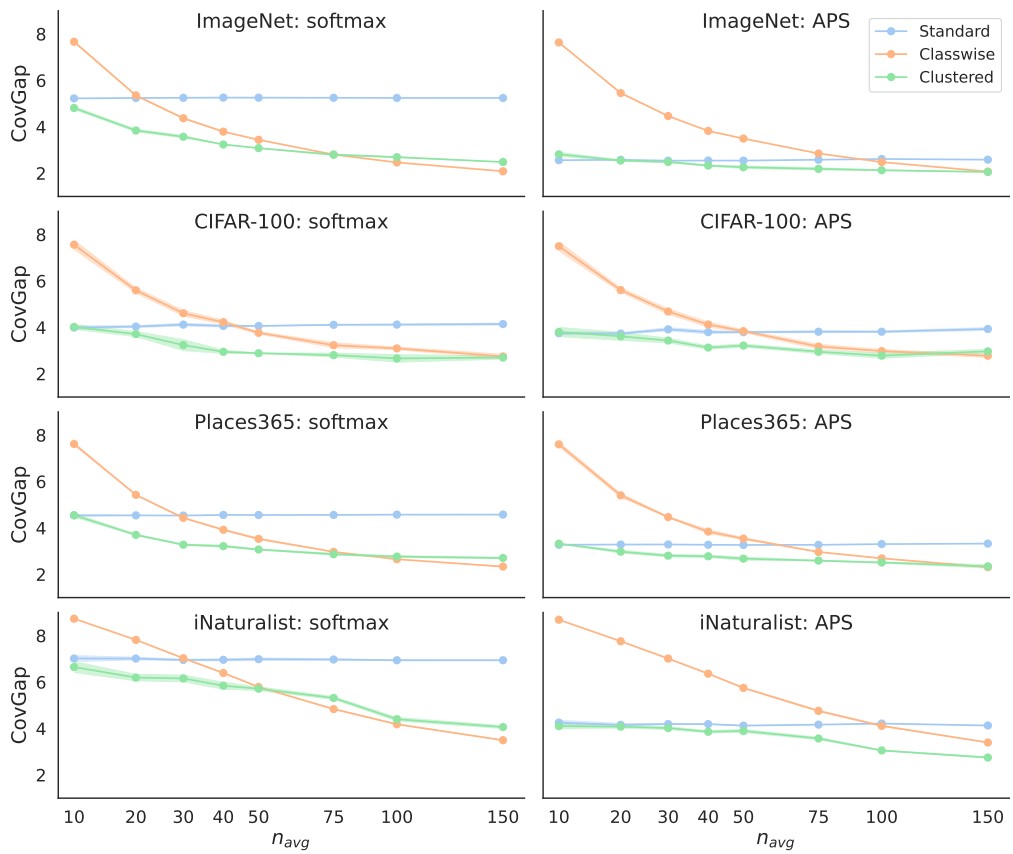

Figure 2: Average class coverage gap for ImageNet, CIFAR-100, Places365, and iNaturalist, for the softmax (left) and APS (right) scores, as we vary the average number of calibration examples per class. The shaded regions denote $\pm 1.96$ times the standard errors (often, the standard errors are too small to be visible).

latter for small enough values of $n_{\text{avg}}$. However, the behavior of STANDARD changes notably as we move from softmax to APS: it becomes comparable to CLUSTERED and only slightly worse for large $n_{\text{avg}}$. Lastly, the results for RAPS (not shown, and deferred to Appendix C.1) are similar to APS but the CovGap is shifted slightly higher.

To examine the potential tradeoffs between class-conditional coverage and average set size, we focus on the $n_{\text{avg}} \in \{10, 20, 50, 75\}$ settings and report CovGap and AvgSize in Table 2 for all data sets and score functions. We see that CLUSTERED achieves the best or near-best CovGap in any experimental combination, and its improvement over the baselines is particularly notable in the regime where data is limited, but not extremely limited ($n_{\text{avg}} = 20$ or 50). Meanwhile, AvgSize for STANDARD and CLUSTERED is relatively stable across values of $n_{\text{avg}}$, with the latter being generally slightly larger; AvgSize for CLASSWISE decreases as we increase $n_{\text{avg}}$, but is still quite large relative to STANDARD and CLUSTERED, especially for iNaturalist.

We finish with two more remarks. First, we find CovGap tends to behave quite similarly to the fraction of classes that are drastically undercovered, which we define as having a class-conditional coverage at least 10% below the desired level. Results for this metric are given in Appendix C.2. Second, as is the case in any conformal method, auxiliary randomization can be applied to STANDARD, CLASSWISE, or CLUSTERED in order to achieve a coverage guarantee (marginal, class-conditional, or cluster-conditional, respectively) of *exactly* $1 - \alpha$, rather than *at least* $1 - \alpha$. These results are included in Appendix C.3. In terms of CovGap, we find that randomization generally improves CLASSWISE, and does not change STANDARD or CLUSTERED much at all; however, the set sizes from randomized CLASSWISE are still overall too large to be practically useful (moreover, injecting auxiliary randomness is prediction sets is not always practically palatable).

## 4   Discussion

We summarize our practical takeaways, in an effort towards creating guidelines for answering the question: *for a given problem setting, what is the best way to produce prediction sets that have good class-conditional coverage but are not too large to be useful?*

- *Extremely low-data regime.* When most classes have very few calibration examples (say, less than 10), this is not enough data to run CLUSTERED or CLASSWISE unless $\alpha$ is large, so the only option is to run STANDARD. With this method, softmax and RAPS are both good score functions. RAPS tends to yield better class-conditional coverage, while softmax tends to have smaller sets.

- *Low-data regime.* When the average number of examples per class is low but not tiny (say, around 20 to 75), CLUSTERED tends to strike a good balance between variance and granularity and often achieves good class-conditional coverage and reasonably-sized prediction sets with either softmax or RAPS. The STANDARD method with RAPS is also competitive towards the lower-data end of this regime.

- *High-data regime.* When the average number of examples per class is large (say, over 75), CLUSTERED conformal with either softmax or RAPS continues to do well, and CLASSWISE with these same scores can also do well if the classes are balanced. In settings with high class imbalance, CLASSWISE is unstable for rare classes and produces excessively large prediction sets, whereas CLUSTERED is more robust due to the data-sharing it employs.

- *Extremely high-data regime.* When the calibration dataset is so large that the rarest class has at least, say, 100 examples, then CLASSWISE with softmax or RAPS will be a good choice, regardless of any class imbalance.

The boundaries between these regimes are not universal and are dependent on the data characteristics, and the above guidelines are based only on our findings from our experiments. We also note that the precise boundary between the low-data and high-data regimes is dependent on the particular score function that is used: the transition happens around 20-40 examples per class for softmax and 50-100 for RAPS. This serves as further motivation for CLUSTERED, which performs relatively well in all regimes.

As a possible direction for future work, it might be interesting to generalize our approach to the broader problem of group-conditional coverage with many groups. In the present setting, the groups

Table 2: Average class coverage gap and average set size for select values of $n_{\text{avg}}$ on ImageNet, CIFAR-100, Places365, and iNaturalist. Bold emphasizes the best (within $\pm 0.2$) class coverage gap in each experimental combination. Standard errors are reported in parentheses.

| Data set | Score | Method | $n_{\text{avg}} = 10$ CovGap | AvgSize | $n_{\text{avg}} = 20$ CovGap | AvgSize | $n_{\text{avg}} = 50$ CovGap | AvgSize | $n_{\text{avg}} = 75$ CovGap | AvgSize |
|---|---|---|---|---|---|---|---|---|---|---|
| ImageNet | softmax | STANDARD | 5.2 (0.0) | 1.9 (0.0) | 5.2 (0.0) | 1.9 (0.0) | 5.3 (0.0) | 1.9 (0.0) | 5.2 (0.0) | 1.9 (0.0) |
| | | CLASSWISE | 7.7 (0.0) | 354.4 (2.0) | 5.4 (0.0) | 23.0 (0.9) | 3.4 (0.0) | 5.1 (0.1) | **2.8** (0.0) | 4.2 (0.1) |
| | | CLUSTERED | **4.9** (0.1) | 2.5 (0.1) | **3.9** (0.0) | 2.7 (0.1) | **3.1** (0.0) | 2.6 (0.1) | **2.8** (0.0) | 2.7 (0.0) |
| | APS | STANDARD | **2.6** (0.0) | 25.9 (0.3) | **2.6** (0.0) | 25.8 (0.1) | **2.5** (0.0) | 25.6 (0.0) | 2.6 (0.0) | 25.7 (0.1) |
| | | CLASSWISE | 7.6 (0.0) | 394.2 (1.9) | 5.5 (0.0) | 76.9 (0.8) | 3.5 (0.0) | 39.7 (0.2) | 2.9 (0.0) | 35.0 (0.2) |
| | | CLUSTERED | 3.0 (0.1) | 29.6 (1.5) | **2.6** (0.0) | 27.0 (0.7) | **2.3** (0.0) | 27.2 (0.5) | **2.2** (0.0) | 27.3 (0.4) |
| | RAPS | STANDARD | **3.0** (0.0) | 5.2 (0.0) | **3.0** (0.0) | 5.1 (0.0) | 3.0 (0.0) | 5.1 (0.0) | 3.0 (0.0) | 5.1 (0.0) |
| | | CLASSWISE | 7.7 (0.1) | 361.9 (1.9) | 5.6 (0.0) | 34.2 (1.0) | 3.4 (0.0) | 8.8 (0.3) | 2.8 (0.0) | 7.3 (0.1) |
| | | CLUSTERED | **3.1** (0.0) | 7.7 (1.0) | **2.9** (0.0) | 6.6 (0.6) | **2.6** (0.0) | 6.5 (0.3) | **2.4** (0.0) | 6.8 (0.3) |
| CIFAR-100 | softmax | STANDARD | **4.0** (0.1) | 8.1 (0.2) | 4.0 (0.0) | 7.9 (0.2) | 4.1 (0.0) | 7.9 (0.1) | 4.1 (0.0) | 8.0 (0.1) |
| | | CLASSWISE | 7.6 (0.1) | 47.2 (0.6) | 5.6 (0.1) | 19.3 (0.5) | 3.8 (0.0) | 11.9 (0.2) | 3.2 (0.1) | 10.8 (0.1) |
| | | CLUSTERED | **4.2** (0.1) | 8.9 (0.5) | **3.6** (0.1) | 8.9 (0.3) | **2.9** (0.1) | 9.3 (0.3) | **2.8** (0.1) | 9.1 (0.2) |
| | APS | STANDARD | **3.7** (0.1) | 11.2 (0.3) | **3.7** (0.0) | 10.8 (0.2) | 3.8 (0.0) | 11.0 (0.1) | 3.8 (0.0) | 11.0 (0.1) |
| | | CLASSWISE | 7.5 (0.1) | 49.4 (0.5) | 5.6 (0.1) | 22.6 (0.5) | 3.8 (0.1) | 15.0 (0.2) | 3.2 (0.1) | 13.7 (0.1) |
| | | CLUSTERED | 4.0 (0.1) | 11.9 (0.5) | **3.6** (0.1) | 12.1 (0.5) | **3.1** (0.1) | 12.2 (0.4) | **2.8** (0.1) | 12.2 (0.2) |
| | RAPS | STANDARD | **4.7** (0.1) | 8.6 (0.3) | 4.8 (0.1) | 8.3 (0.3) | 4.8 (0.0) | 8.3 (0.1) | 4.9 (0.1) | 8.1 (0.1) |
| | | CLASSWISE | 7.5 (0.1) | 47.5 (0.7) | 5.6 (0.1) | 20.7 (0.6) | **3.8** (0.1) | 13.3 (0.2) | **3.3** (0.1) | 12.1 (0.1) |
| | | CLUSTERED | **4.9** (0.1) | 9.5 (0.5) | **4.5** (0.1) | 8.7 (0.5) | **3.7** (0.1) | 9.5 (0.3) | **3.4** (0.1) | 9.9 (0.1) |
| Places365 | softmax | STANDARD | **4.5** (0.0) | 6.9 (0.1) | 4.5 (0.0) | 6.9 (0.0) | 4.6 (0.0) | 6.9 (0.0) | 4.6 (0.0) | 6.9 (0.0) |
| | | CLASSWISE | 7.6 (0.1) | 136.5 (1.7) | 5.4 (0.0) | 18.0 (0.2) | 3.5 (0.0) | 10.1 (0.1) | **3.0** (0.0) | 9.3 (0.1) |
| | | CLUSTERED | **4.5** (0.1) | 7.1 (0.1) | **3.9** (0.1) | 7.1 (0.1) | **3.0** (0.0) | 7.8 (0.1) | **2.9** (0.1) | 7.9 (0.1) |
| | APS | STANDARD | **3.3** (0.0) | 10.8 (0.1) | **3.3** (0.0) | 10.9 (0.1) | 3.3 (0.0) | 10.8 (0.1) | 3.3 (0.0) | 10.8 (0.1) |
| | | CLASSWISE | 7.6 (0.1) | 140.5 (1.7) | 5.4 (0.1) | 23.5 (0.3) | 3.5 (0.0) | 14.4 (0.1) | 3.0 (0.0) | 13.4 (0.1) |
| | | CLUSTERED | 3.5 (0.1) | 10.9 (0.3) | **3.1** (0.0) | 10.9 (0.2) | **2.6** (0.0) | 11.8 (0.1) | **2.7** (0.1) | 11.6 (0.2) |
| | RAPS | STANDARD | **3.8** (0.0) | 7.9 (0.0) | 3.9 (0.0) | 7.9 (0.0) | 3.8 (0.0) | 7.9 (0.0) | 3.8 (0.0) | 7.9 (0.0) |
| | | CLASSWISE | 7.6 (0.1) | 138.9 (1.7) | 5.5 (0.1) | 21.3 (0.3) | 3.6 (0.0) | 11.4 (0.1) | **3.0** (0.0) | 10.5 (0.1) |
| | | CLUSTERED | **3.9** (0.1) | 8.0 (0.1) | **3.4** (0.0) | 8.2 (0.2) | **3.0** (0.0) | 8.8 (0.1) | **2.9** (0.0) | 9.0 (0.2) |
| iNaturalist | softmax | STANDARD | **7.0** (0.1) | 3.1 (0.0) | 7.0 (0.1) | 3.1 (0.0) | 7.0 (0.1) | 3.2 (0.0) | 7.0 (0.0) | 3.2 (0.0) |
| | | CLASSWISE | 8.7 (0.0) | 469.4 (1.8) | 7.8 (0.0) | 364.0 (1.4) | **5.8** (0.0) | 148.7 (2.2) | **4.8** (0.0) | 55.3 (1.6) |
| | | CLUSTERED | **6.9** (0.2) | 3.1 (0.1) | **6.4** (0.1) | 3.4 (0.1) | **5.7** (0.1) | 3.7 (0.0) | 5.3 (0.1) | 3.8 (0.0) |
| | APS | STANDARD | **4.3** (0.1) | 8.3 (0.1) | **4.2** (0.1) | 8.4 (0.1) | **4.1** (0.0) | 8.3 (0.0) | 4.2 (0.0) | 8.3 (0.0) |
| | | CLASSWISE | 8.7 (0.0) | 472.8 (1.8) | 7.8 (0.0) | 368.4 (1.4) | 5.7 (0.0) | 153.9 (2.2) | 4.8 (0.0) | 60.7 (1.5) |
| | | CLUSTERED | **4.3** (0.1) | 8.1 (0.1) | **4.1** (0.1) | 8.3 (0.1) | **3.9** (0.0) | 8.5 (0.1) | **3.5** (0.0) | 8.7 (0.1) |
| | RAPS | STANDARD | **5.1** (0.1) | 5.1 (0.0) | **5.0** (0.0) | 5.2 (0.0) | 5.0 (0.0) | 5.1 (0.0) | 5.0 (0.0) | 5.2 (0.0) |
| | | CLASSWISE | 8.7 (0.0) | 473.8 (1.9) | 7.8 (0.0) | 369.2 (1.5) | 5.8 (0.0) | 155.3 (2.3) | 4.9 (0.0) | 63.1 (1.7) |
| | | CLUSTERED | **5.0** (0.1) | 5.0 (0.0) | **4.9** (0.1) | 5.2 (0.0) | **4.5** (0.1) | 5.4 (0.1) | **4.1** (0.1) | 5.4 (0.1) |

are defined by class labels, but our clustering methodology could also be applied to other group structures (e.g., defined by the input features or components of a mixture distribution).

## Acknowledgments and Disclosure of Funding

We thank Margaux Zaffran and Long Nguyen for helpful suggestions that improved our paper. This work was supported by the National Science Foundation (NSF) Graduate Research Fellowship Program under grant no. 2146752, the European Research Council (ERC) Synergy Grant Program, the Office of Naval Research (ONR) the Mathematical Data Science Program, and the ONR Multi-University Research Initiative (MURI) Program under grant no. N00014-20-1-2787.

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

## A Proofs

**Proof of Proposition 1.** For each $m \in \{1, \ldots, M\}$, denote by $G^{(m)}$ the distribution of the score $s(X, Y)$ conditioned on $Y$ being in cluster $m$. Consider a randomly sampled test example $(X_{\text{test}}, Y_{\text{test}})$ with a label in cluster $m$; the test score $s_{\text{test}} = s(X_{\text{test}}, Y_{\text{test}})$ then follows distribution $G^{(m)}$. Next consider $\{s_i\}_{i \in \mathcal{I}_2(m)}$, the scores for examples in the proper calibration set with labels in cluster $m$; these also follow distribution $G^{(m)}$. Furthermore, $s_{\text{test}}$ and the elements of $\{s_i\}_{i \in \mathcal{I}_2(m)}$ are all mutually independent, so the result follows by the standard coverage guarantee for conformal prediction with exchangeable scores (e.g., see Vovk et al. (2005) or Lei et al. (2018)).

**Proof of Proposition 2.** This is a direct result of exchangeability and Proposition 1.

**Proof of Proposition 3.** Let $S = s(X, Y)$ denote the score of a randomly sampled example $(X, Y) \sim F$. Fix an $m \in \{1, \ldots, M\}$. Define $\mathcal{Y}^{(m)} = \{y \in \mathcal{Y} : \hat{h}(y) = m\}$ as the set of classes that $\hat{h}$ assigns to cluster $m$. Without a loss of generality, we treat both $\hat{h}$ and $\hat{q}(m)$ as fixed for the remainder of this proof. This can be done by conditioning on both the clustering and proper calibration sets, leaving only the randomness in the test point $(X, Y) \sim F$, and then integrating over the clustering and proper calibration sets in the end.

Let $G^{(m)}$ denote the distribution of $S$ conditional on $Y \in \mathcal{Y}^{(m)}$, and let $S^{(m)} \sim G^{(m)}$. Similarly, let $G^y$ denote the distribution of $S$ conditional on $Y = y$, and let $S^y \sim G^y$. Since we assume that the KS distance between the score distribution for every pair of classes in cluster $m$ is bounded by $\epsilon$, and $G^{(m)}$ is a mixture of these distributions (that is, $G^{(m)} = \sum_{y \in \mathcal{Y}^{(m)}} \pi_y \cdot G^y$ for some fixed probability weights $\pi_y, y \in \mathcal{Y}^{(m)}$), it follows by the triangle inequality that

$$D_{\text{KS}}(S^y, S^{(m)}) \leq \epsilon, \quad \text{for all } y \in \mathcal{Y}^{(m)}.$$

By definition of KS distance, this implies

$$\left| \mathbb{P}(S \leq \hat{q}(m) \mid Y = y) - \mathbb{P}(S \leq \hat{q}(m) \mid Y \in \mathcal{Y}^{(m)}) \right| \leq \epsilon.$$

Since the CLUSTERED procedure includes the true label $Y$ from the prediction set $C$ when $S \leq \hat{q}(m)$, these probabilities can be rewritten in terms of coverage events:

$$\left| \mathbb{P}(Y \in \mathcal{C}(X) \mid Y = y) - \mathbb{P}(Y \in \mathcal{C}(X) \mid Y \in \mathcal{Y}^{(m)}) \right| \leq \epsilon.$$

Combining the result from Proposition 1 gives the desired conclusion.

## B Experiment details

### B.1 Score functions

We perform experiments using three score functions:

- softmax: The softmax-based conformal score at an input $x$ and a label $y$ is defined as

$$s_{\text{softmax}}(x, y) = 1 - f_y(x),$$

  where $f_y(x)$ is entry $y$ of the softmax vector output by the classifier $f$ at input $x$.

- APS: The *Adaptive Prediction Sets* (APS) score of Romano et al. (2020b) is designed to improve $X$-conditional coverage as compared to the more traditional softmax score. This score is computed at an input $x$ and label $y$ as follows. Let

$$f_{(1)}(x) \leq f_{(2)}(x) \leq \cdots \leq f_{(|\mathcal{Y}|)}(x)$$

  denote the sorted values of the base classifier softmax outputs $f_y(x), y \in \mathcal{Y}$. Let $k_x(y)$ be the index in the sorted order that corresponds to class $y$, that is, $f_{(k_x(y))} = f_y(x)$. The APS score is then defined as

$$s_{\text{APS}}(x, y) = \sum_{i=1}^{k_x(y)-1} f_{(i)}(x) + \text{Unif}([0, f_{(k_x(y))}(x)]).$$

- RAPS: The *regularized APS* (RAPS) score of Angelopoulos et al. (2021) is a modification of the APS score that adds a regularization term designed to reduce the prediction set sizes (which can often be very large with APS). The RAPS score is defined as

$$s_{\mathsf{RAPS}}(x, y) = s_{\mathsf{APS}}(x, y) + \max(0, \lambda(k_x(y) - k_{\mathrm{reg}})),$$

where $k_x(y)$ is as defined above, and $\lambda$ and $k_{\mathrm{reg}}$ are user-chosen parameters. In all of our experiments, we use $\lambda = 0.01$ and $k_{\mathrm{reg}} = 5$, which Angelopoulos et al. (2021) found to work well for ImageNet.

## B.2 Model training

An important consideration when we fine-tune and calibrate our models is that we must reserve sufficient data to evaluate the class-conditional coverage of the conformal methods. This means we aim to exclude at least 250 examples per class from the fine-tuning and calibration sets so that we can then use this data for validation (applying the conformal methods and computing coverage and set size metrics).

For all data sets except ImageNet, we use a ResNet-50 model as our base classifier. We initialize to the `IMAGENET1K_V2` pre-trained weights from `PyTorch` (Paszke et al., 2019), and then fine-tune all parameters by training on the data set at hand. For ImageNet, we must do something different, as explained below.

**ImageNet.**   Setting up this data set for our experiments is a bit tricky because we need sufficient data for performing validation, but we also need this data to be separate from the fine-tuning and calibration sets. The ImageNet validation set only contains 50 examples per class, which is not enough for our setting. The ImageNet training set is much larger, with roughly 1000 examples per class, but if we want to use this data for validation, then we cannot use the ResNet-50 initialized to the `IMAGENET1K_V2` pre-trained weights, as these weights were obtained by training on the whole ImageNet training set. We therefore instead use a SimCLR-v2 model (Chen et al., 2020), which was trained on the ImageNet training set *without labels*, to extract feature vectors of length 6144 for all images in the ImageNet training set. We then use 10% of these feature vectors for training a linear head (a single fully connected neural network layer). After training for 10 epochs, the model achieves a validation accuracy of 78%. We then apply the linear head to the remaining 90% of the feature vectors to obtain softmax scores for the calibration set.

**CIFAR-100.**   This data set has 600 images per class (500 from the training set and 100 from the validation set). We combine the data and then randomly sample 50% for fine-tuning, and we use the remaining data for calibrating and validating our procedures. After training for 30 epochs, the validation accuracy is 60%.

**Places365.**   This data set contains more than 10 million images of 365 classes, where each class has 5000 to 30000 examples. We randomly sample 90% of the data for fine-tuning, and we use the remaining data for calibrating and validating our procedures. After training for one epoch, the validation accuracy is 52%.

**iNaturalist.**   This data set has class labels of varying specificity. For example, at the `species` level, there are 6414 classes with 300 examples each (290 training examples and 10 validation examples) and a total of 10000 classes with at least 150 examples. We instead work at the `family` level, which groups the species into 1103 classes. We randomly sample 50% of the data for fine-tuning, and we use the remaining for calibrating and validating our procedures. After training for one epoch, the validation accuracy is 69%.

However, due to high class imbalance and sampling randomness, some classes have insufficient validation samples, so we filter out classes with fewer than 250 validation examples, which leaves us with 633 classes. The entries of the softmax vectors that correspond to rare classes are removed and the vector is renormalized to sum to one.

### B.3 Measuring class balance in Table 1

The class balance metric in Table 1 is defined as the number of examples in the rarest 5% of classes divided by the expected number of examples if the class distribution were perfectly uniform. This metric is bounded between 0 and 1, with lower values denoting more class imbalance. We compute this metric using $D_{\text{fine}}^c$.

### B.4 Choosing clustering parameters

To choose $\gamma \in [0, 1]$ and $M \geq 1$ for CLUSTERED, as described briefly in Section 3.1, we employ two intuitive heuristics. We restate these heuristics in more detail here.

- First, to distinguish between more clusters (or distributions), we need more samples from each distribution. As a rough guess, to distinguish between two distributions, we want at least four samples per distribution; to distinguish between five distributions, we want at least ten samples per distribution. In other words, we want the number of clustering examples per class to be at least twice the number of clusters. This heuristic can be expressed as

$$\gamma \tilde{n} \geq 2M, \tag{3}$$

  where $\gamma \tilde{n}$ is the expected number of clustering examples for the rarest class that is not assigned to the null cluster.

- Second, we want enough data for computing the conformal quantiles for each cluster. Specifically, we seek at least 150 examples per cluster on average. This heuristic can be expressed as

$$(1 - \gamma)\tilde{n}\frac{K}{M} \geq 150, \tag{4}$$

  where $K/M$ is the average number of classes per cluster and $(1 - \gamma)\tilde{n}$ is the expected number of proper calibration examples for the rarest class not assigned to the null cluster.

Changing the inequalities of (3) and (4) into equalities and solving for $\gamma$ and $M$ yields

$$M = \frac{\gamma \tilde{n}}{2} \qquad \text{and} \qquad \gamma = \frac{K}{K + 75}.$$

**Varying the clustering parameters.** As sensitivity analysis, we examine the performance of CLUSTERED across a wide range of values for the tuning parameters $\gamma$ and $M$. As the heatmaps in Figure 3 confirm, the performance of CLUSTERED is not particularly sensitive to the values of these parameters. When $n_{\text{avg}} = 10$, the heuristic chooses $\gamma = 0.89$ and $M = 4$, meanwhile, when $n_{\text{avg}} = 50$, the heuristic chooses $\gamma \in [0.88, 0.92]$ and $M \in [7, 12]$ (since the calibration data set is randomly sampled, and $\gamma$ and $M$ are chosen based on the calibration data set, there is randomness in the chosen values). However, there are large areas surrounding these values that would yield similar performance. We observe that the heuristics do not always choose the parameter values that yield the lowest CovGap. The heatmaps reveal that the optimal parameter values are dependent not only on data set characteristics, but also on the score function. Future work could be done to extract further performance improvements by determining a better method for choosing $\gamma$ and $M$.

## C  Additional experimental results

We present additional experimental results in this section. As in the main text, the shaded regions in plots denote $\pm 1.96$ times the standard errors.

### C.1  RAPS **CovGap results**

Figure 4 shows the CovGap on all data sets when we use RAPS as our score function.

### C.2  Additional metrics

**Average set size.** To supplement Table 2 from the main text, which reports AvgSize for select values of $n_{\text{avg}}$, Figure 5 plots AvgSize for all values of $n_{\text{avg}}$ that we use in our experimental setup. Note

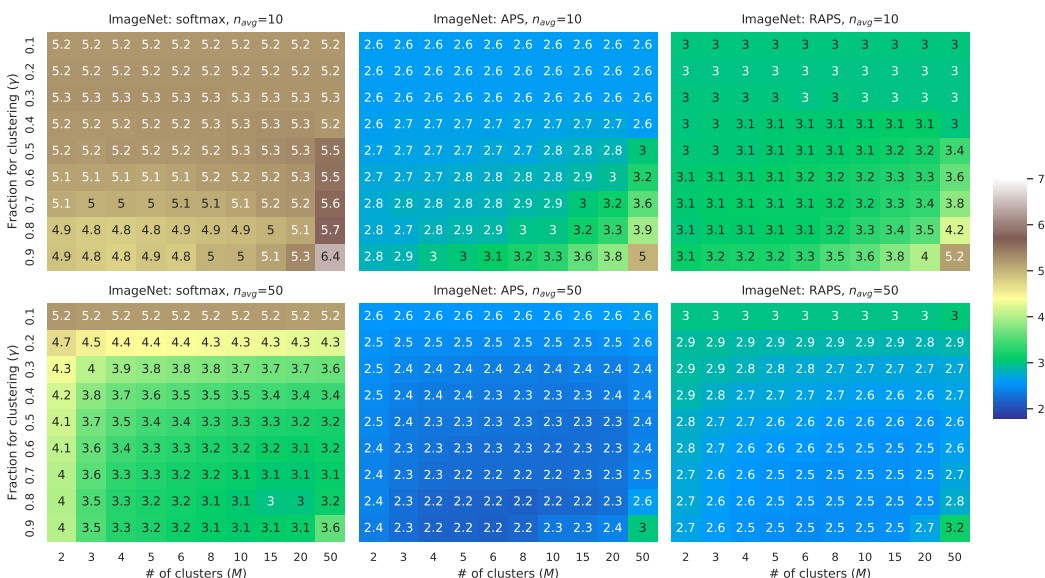

Figure 3: Average class coverage gap on ImageNet for $n_{\mathrm{avg}} \in \{10, 50\}$, using the softmax, APS, and RAPS scores, as we vary the clustering parameters. Each entry is averaged across 10 random splits of the data into calibration and validation sets.

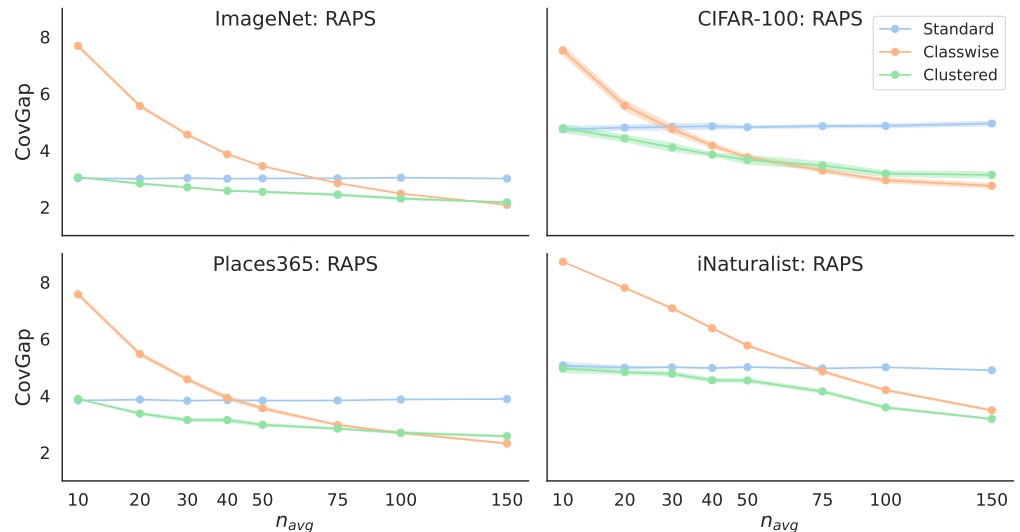

Figure 4: Average class coverage gap for ImageNet, CIFAR-100, Places365, and iNaturalist, using the RAPS score, as we vary the average number of calibration examples per class.

that RAPS sharply reduces AvgSize relative to APS on ImageNet and also induces a slight reduction for the other three data sets. This asymmetric reduction is likely due to the fact that the RAPS hyperparameters, which control the strength of the set size regularization, were tuned on ImageNet. The set sizes of RAPS on other data sets could likely be improved by tuning the hyperparameters for each data set.

**Fraction undercovered.** In various practical applications, we want to limit the number of classes that are severely undercovered, which we define as having a class-conditional coverage more than 10% below the desired coverage level. We define the fraction of undercovered classes (FracUnderCov)

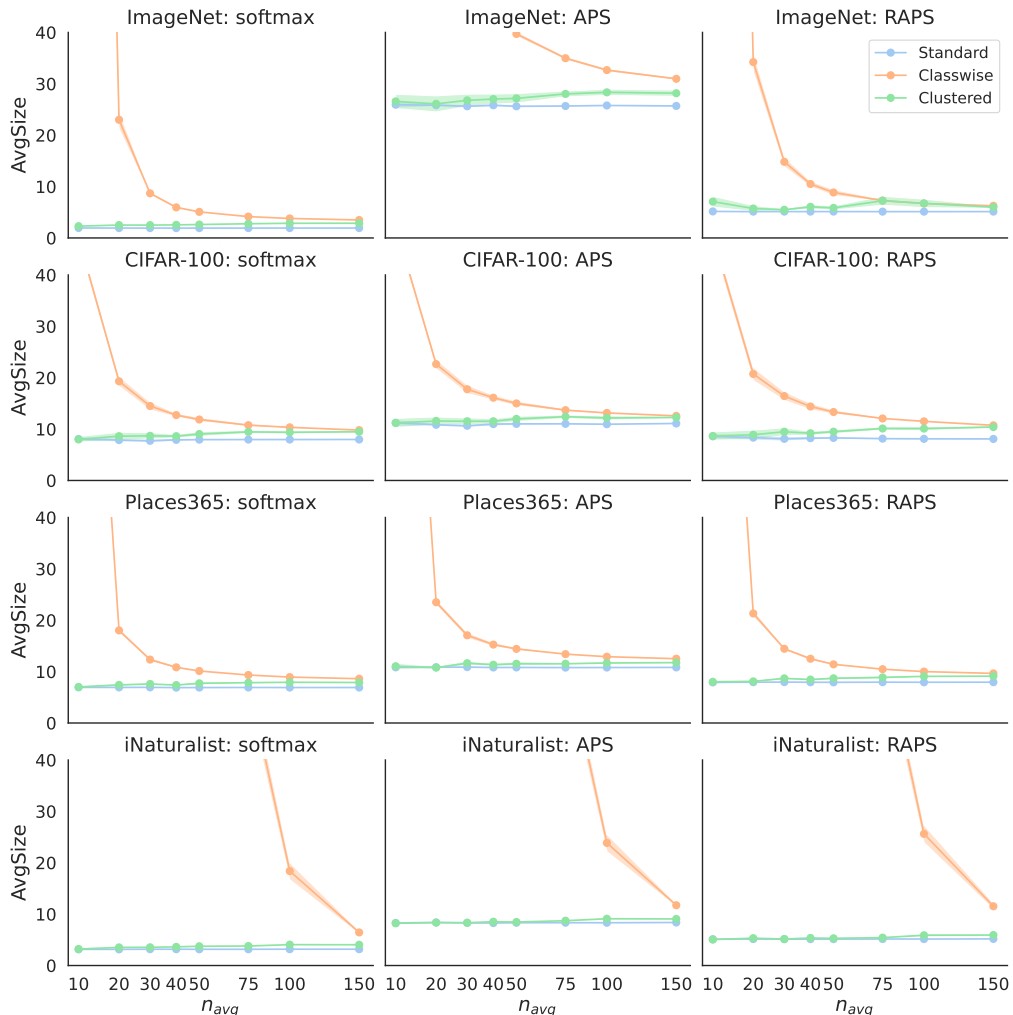

Figure 5: Average set size for ImageNet, CIFAR-100, Places365, and iNaturalist, for the softmax, APS, and RAPS scores, as we vary the average number of calibration examples per class.

metric as:

$$\text{FracUnderCov} = \frac{1}{|\mathcal{Y}|} \sum_{y=1}^{|\mathcal{Y}|} \mathbb{1}\left\{ \hat{c}_y \leq 1 - \alpha - 0.1 \right\},$$

recalling that $\hat{c}_y$ is the empirical class-conditional coverage for class $y$. Figure 6 plots FracUnderCov for all experimental settings. Comparing to the CovGap plots in Figure 2 and Figure 4, we see that the trends in FracUnderCov generally mirror the trends in CovGap. However, FracUnderCov is a much noisier metric, as evidenced by the large error bars. Another difference is that FracUnderCov as a metric is unable to penalize uninformatively large set sizes. This is best seen in the performance of CLASSWISE on iNaturalist: for every score function, CLASSWISE has very low FracUnderCov, but this is achieved by producing extremely large prediction sets, as shown in the bottom row of Figure 5. Meanwhile, CovGap is somewhat able to penalize this kind of behavior since unnecessarily large set sizes often lead to overcoverage, and CovGap penalizes overcoverage.

### C.3 Auxiliary randomization

The conformal methods in the main paper generate *deterministic* prediction sets, so running the method on the same input will always produce the same prediction set. These prediction sets are designed to achieve *at least* $1 - \alpha$ marginal, class-conditional, or cluster-conditional coverage. In

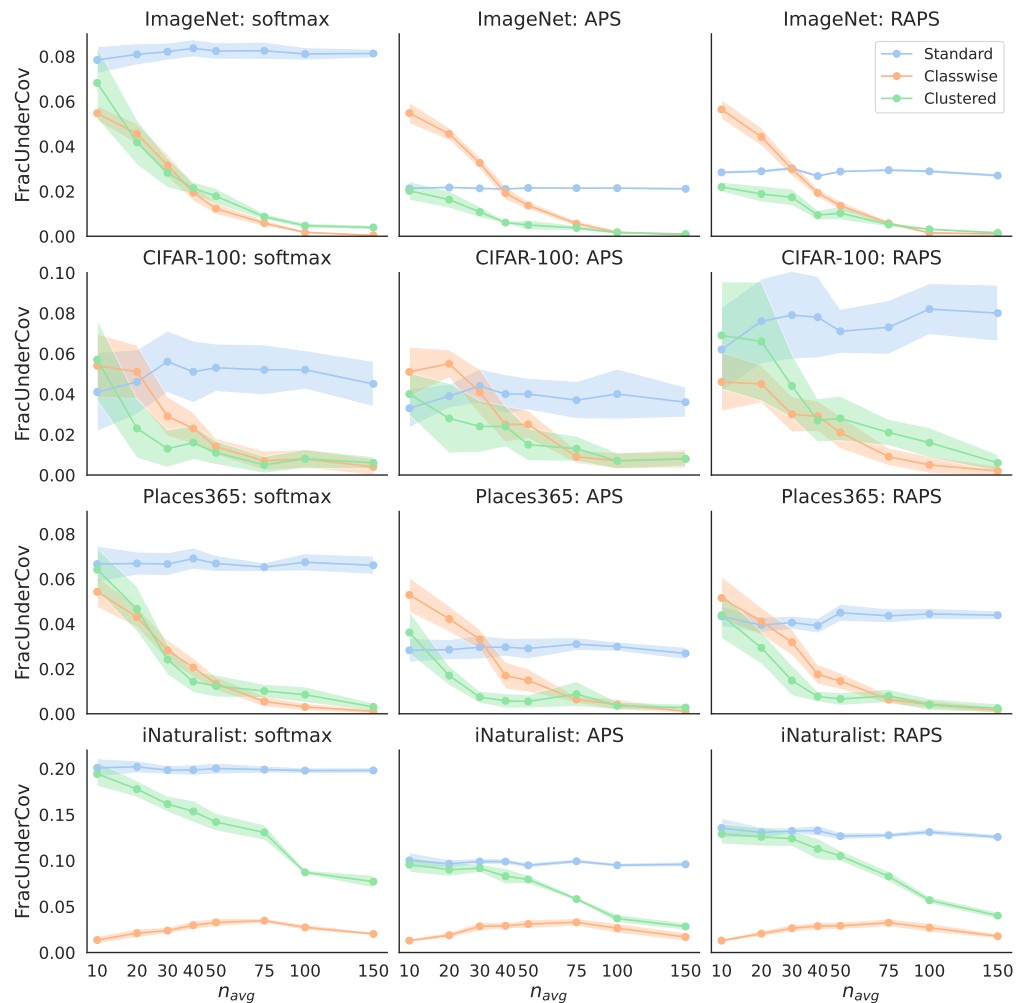

Figure 6: Fraction of severely undercovered classes for ImageNet, CIFAR-100, Places365, and iNaturalist, using the softmax, APS, and RAPS scores, as we vary the average number of calibration examples per class.

most practical situations, it is arguably undesirable to use non-deterministic or *randomized* prediction sets (say, if you are a patient, you would not want your doctor to tell you that your diagnoses change depending on a random seed). However, if one is willing to accept randomized prediction sets, then the conformal methods described in the main text can be modified to achieve *exact* $1 - \alpha$ coverage.

**Randomizing to achieve exact $1 - \alpha$ coverage.** Recall that the unrandomized STANDARD method used in the main paper uses

$$\hat{q} = \text{Quantile}\left(\frac{\lceil (N+1)(1-\alpha) \rceil}{N}, \{s_i\}_{i=1}^N\right),$$

which yields a coverage guarantee of

$$\mathbb{P}(Y_\text{test} \in \mathcal{C}(X_\text{test}) \mid Y_\text{test} = y) = \frac{\lceil (N+1)(1-\alpha) \rceil}{N+1} \geq 1 - \alpha,$$

under the assumption that the scores are distinct almost surely. The equality above holds because the event that $Y_\text{test}$ is included in $\mathcal{C}(X_\text{test})$ is equivalent to the event that $s(X_\text{test}, Y_\text{test})$ is one of the $\lceil (N+1)(1-\alpha) \rceil$ smallest scores in the set containing the calibration points and the test point, and, by exchangeability, this occurs with probability exactly $\lceil (N+1)(1-\alpha) \rceil / (N+1)$. By similar

reasoning, if we instead use

$$\tilde{q} = \mathrm{Quantile}\left(\frac{\lceil (N+1)(1-\alpha) \rceil - 1}{N}, \{s_i\}_{i=1}^N\right)$$

as our conformal quantile (note the added $-1$ in the numerator), then we would have

$$\mathbb{P}(Y_{\mathrm{test}} \in \mathcal{C}(X_{\mathrm{test}}) \mid Y_{\mathrm{test}} = y) = \frac{\lceil (N+1)(1-\alpha) \rceil - 1}{N+1} < 1 - \alpha.$$

To summarize, using $\hat{q}$ results in at least $1 - \alpha$ coverage, and using $\tilde{q}$ results in less than $1 - \alpha$ coverage. Thus, to achieve exact $1 - \alpha$ coverage, we can randomize between using $\hat{q}$ and $\tilde{q}$. Let

$$b = \frac{\lceil (N+1)(1-\alpha) \rceil}{N+1} - (1 - \alpha)$$

be the amount by which the coverage using $\hat{q}$ overshoots the desired coverage level and let

$$c = (1 - \alpha) - \frac{\lceil (N+1)(1-\alpha) \rceil - 1}{N+1}$$

be the amount by which the coverage using $\tilde{q}$ undershoots the desired coverage level. Then, if we define the Bernoulli random variable

$$B \sim \mathrm{Bern}\left(\frac{c}{b+c}\right),$$

independent of everything else that is random, and set

$$\hat{q}_{\mathrm{rand}} = B\hat{q} + (1 - B)\tilde{q}$$

then the prediction sets created using $\hat{q}_{\mathrm{rand}}$ will have exact $1 - \alpha$ marginal coverage. The same idea translates to CLASSWISE and CLUSTERED methods (where we randomize $\hat{q}^y$ and $\hat{q}(m)$, respectively).

Figures 7, 8, and 9 display the CovGap, AvgSize, and FracUnderCov for the randomized versions of the conformal methods. Comparing against earlier plots, we observe that the performance of unrandomized and randomized STANDARD and CLUSTERED are essentially identical in terms of all three metrics. However, we cam see that randomized CLASSWISE exhibits a large improvement relative to unrandomized CLASSWISE in terms of CovGap and AvgSize. That said, the previously-given qualitative conclusions do not change, and the set sizes of randomized CLASSWISE are still too large to be practically useful.

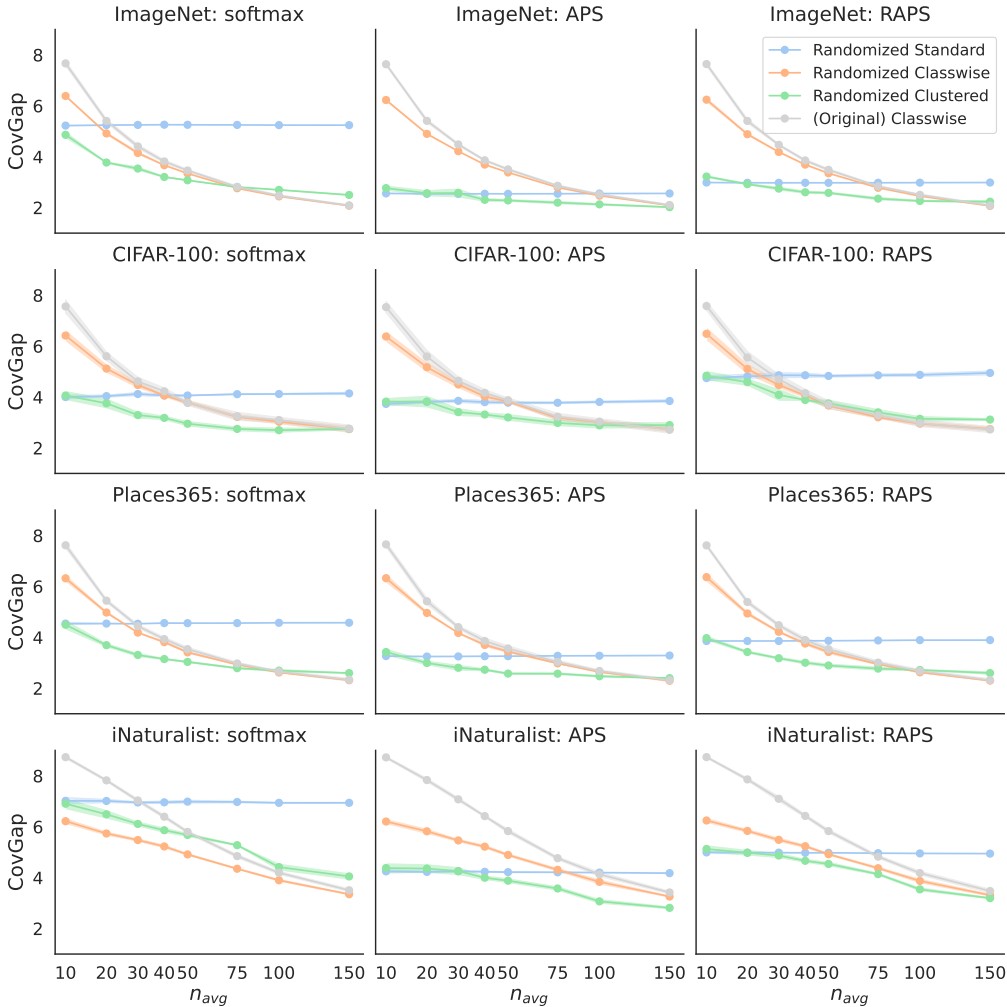

Figure 7: Average class coverage gap for randomized methods on ImageNet, CIFAR-100, Places365, and iNaturalist, for the softmax, APS, and RAPS scores, as we vary the average number of calibration examples per class. The unrandomized original CLASSWISE method is also plotted for comparison purposes.

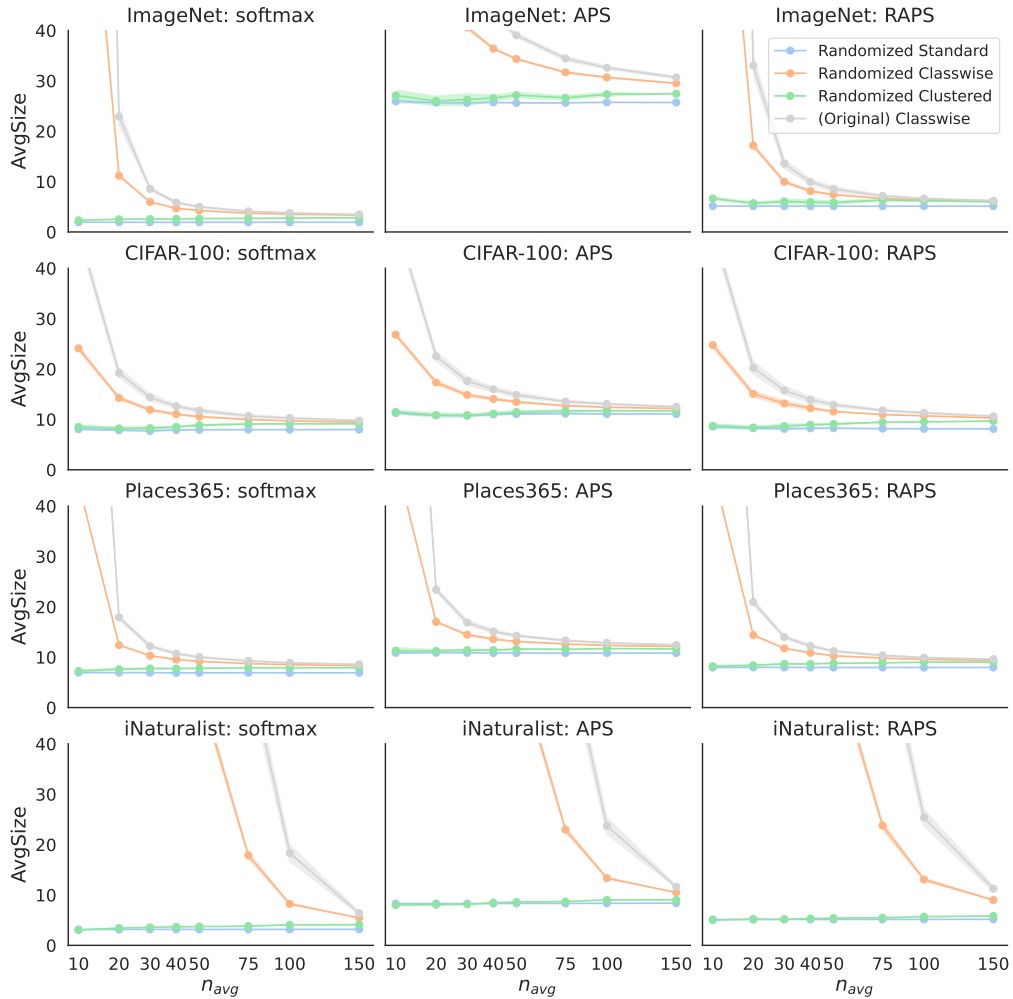

Figure 8: Average set size for randomized methods on ImageNet, CIFAR-100, Places365, and iNaturalist, for the softmax, APS, and RAPS scores, as we vary the average number of calibration examples per class. The unrandomized original CLASSWISE method is also plotted for comparison purposes.

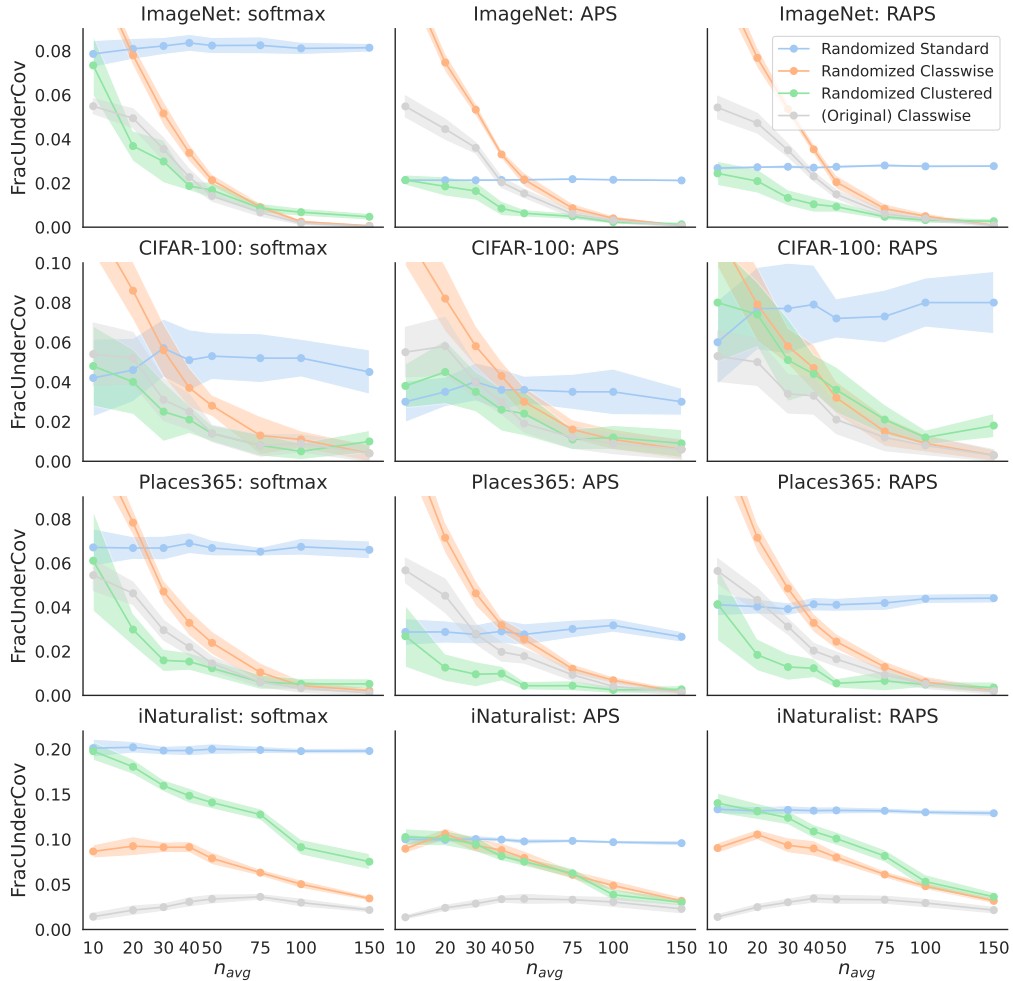

Figure 9: Fraction of severely undercovered classes for randomized methods on ImageNet, CIFAR-100, Pla- ces365, and iNaturalist, for the softmax, APS, and RAPS scores, as we vary the average number of calibration examples per class. The unrandomized original CLASSWISE method is also plotted for comparison purposes.

