# OpenReview forum: "Class-Conditional Conformal Prediction with Many Classes"
_NeurIPS.cc/2023/Conference — NeurIPS 2023 poster_

### Official Review · Reviewer_MBi5 · 2023-07-02

**Soundness:** 2 fair
**Presentation:** 3 good
**Contribution:** 2 fair
**Rating:** 5
**Confidence:** 4

**Summary:**

Standard conformal prediction provides a marginal coverage guarantee which is insufficient for many practical applications. Class-conditional conformal prediction is suitable for many applications, especially in $Y \rightarrow X$ settings, e.g., image data. Achieving class-conditional coverage relies on learning a separate threshold for each label, and hence splitting the calibration set into label-dependent groups. This becomes prohibitive in classification problems with many classes. The current paper proposes a method based on clustering similar classes (which share similar conformal scores) and performing cluster-conditional conformal threshold calibration.

**Strengths:**

The paper is well-written, and provided figures help the reader to follow the content easily. Relevant works and the limitations of the respective methods are highlighted. The proposed method is validated on a large number of large-scale public datasets, illustrating its general practical relevance.

**Weaknesses:**

To avoid tautology, most of my concerns will be outlined in the questions box: those are related to the issues arising empty prediction sets and clustering based on the nonconformity scores. There is also a couple of minor typos:
1. Line 26: $\mathcal{C}(X_{test})\subseteq \mathcal{Y}$.
2. Proposition 1: missing full stop in the equation between lines 156 and 157.

Finally, a couple of stylistic suggestions:
1. In Figure 1, it may be better to make sub-captions consistent (i.e., point out that the left su-plot refers to standard CP and the right one to Classwise CP).
2. In Figure 2, it may be better to use different markers (in addition to different colors) for depicting the results of using different methods.
3. I believe that [1] should also be cited and discussed.

[1] "Least Ambiguous Set-Valued Classifiers with Bounded Error Levels". Sadinle et al., 2016


**Questions:**

I hope that the authors can provide answers to the following questions/concerns:

1. One of the issues with class-conditional CP which has been pointed out in [1] is that there could be cases when the resulting prediction sets are empty (the authors also proposed several ways to handle this issue). Does this issue also apply to the proposed method?

2. I am concerned regarding clustering based on nonconformity scores. Is it possible to (empirically) demonstrate that (even in the most straightforward cases) such classes form "meaningful"/"interpretable" clusters if nonconformity scores are used as features? Consider CIFAR-100 dataset as an example: class "apples" is "closer" to the class "mushrooms" than it is to the class "rocket". However, suppose those happened to be assigned to the same cluster. In that case, the coverage guarantee reads as "among images with apples, mushrooms, and rockets, the true label is contained in the prediction set with probability at least $1-\alpha$" and may not be that interpretable.

3. For many datasets, there is a known taxonomy/hierarchy of classes, e.g., for CIFAR-100, even though there are 100 classes, these are grouped into 20 superclasses. Such taxonomies can also be used to convert the low-data class-conditional calibration step to a high-data group-conditional calibration step, while resulting in interpretable clusters/groups. Even if such taxonomy is not provided, one can still try to come up with one that guarantees that there are "enough" calibration points in each group. What are the advantages of the proposed method over this strategy?

[1] "Least Ambiguous Set-Valued Classifiers with Bounded Error Levels". Sadinle et al., 2016

**Limitations:**

The current work is a part of methodological research and does not have a potential negative societal impact. Overall, I hope that the authors can better highlight the downsides of the proposed method (e.g., grouping based on existing dataset taxonomy).

---

> ### Author Rebuttal · Authors · 2023-08-09
>
> Thank you for your review! We appreciate your summary of the motivation for achieving class-conditional coverage and why it can be difficult in multi-class classification problems. Thanks for suggesting we cite Sadinle et al., 2016, which presents a variant of classwise conformal. We have added it to our discussion of related work.
>
> In response to your questions:
>
> > Q1: Empty sets
>
> In our experiments, we did not observe null sets to be a problem. The problem of empty sets is a real one that appears with some conformal prediction methods, but this problem is orthogonal to the problem we are focused on. If null sets were to appear when the clustered conformal procedure is applied to other settings, we could apply some method for addressing empty sets in other conformal procedures. For example, a simple modification that can be applied to any conformal procedure to avoid empty sets is to return the union of the conformal set and $\\{y_{\min}\\}$ where $y_{\min}$ is the class with the smallest conformal score (i.e., the “most likely” class). Since this modified set is a superset of the original conformal set, the new set will also have a $1-\alpha$ coverage guarantee. (Also, there are further improvements that can make this procedure non-conservative as well.)
>
> In a different direction, Guan and Tibshirani ‘22 [1] use empty sets intentionally and give them semantic meaning. In that work, the empty set is used to indicate that a data point does not appear to be consistent with any class and should be treated as an outlier. While we do not pursue this approach in our present work, with some conformity scores, the same idea could be used in conjunction with the clustering we propose.
>
> [1]  "Prediction and outlier detection in classification problems." Guan and Tibshirani, 2022.
>
> > Q2: Meaningful clusters
>
> Our investigations into cluster memberships showed that indeed, classes that are semantically similar are generally not grouped together (i.e., did not have similar conformal score distributions). If semantically similar classes did have similar conformal score distributions, our problem would be simpler to solve! The fact that this relationship does _not_ hold is what makes our data-driven clustering approach necessary and what prompted our investigation in the first place. Nonetheless, exploring what determines which classes are grouped together is something we would like to better understand. We thank the reviewer for this point and we will explicitly discuss this in the updated manuscript.
>
> > Q3: Clustering based on taxonomy
>
> This is another very interesting direction. The advantage of the alternative method you described is that the clusters are more interpretable, so the cluster-conditional coverage guarantee is perhaps more intuitive. The disadvantage is that this alternative method will not in general yield class-conditional coverage – the errors will be traded off, with some hard classes having low coverage and some easy classes having high coverage.
>
> Conversely, our method produces clusters that are not generally not semantically meaningful, so the cluster-conditional coverage guarantee is less interpretable; however, the advantage is that our method is designed to yield class-conditional coverage (due to the approximate exchangeability of the conformal scores for classes grouped in the same cluster). Echoing our reply to your previous question, the fact that semantically related classes do not have similar distributions is what makes our proposal necessary, and motivated us to pursue this direction in the first place. To summarize: if you want good coverage _for each taxonomic group_, the method you described will be better, but if you want good coverage _for each class_, our proposed method will be better.

---

> > ### Comment · Reviewer_MBi5 · 2023-08-12
> >
> > I thank the authors for their responses to the questions. I have checked the questions/concerns of other reviewers and the corresponding responses, and the experiments with new evaluation metrics. I had a question about the mentioned definition of UnderCovGap: is $|\mathcal{Y}|$ in the denominator correct? Shouldn't it be the sum of indicators (e.g., $\sum_{y\in\mathcal{Y}}1\\{\hat{c}_{y}\leq 1-\alpha\\}$)? (Experiments also suggest so)

---

> > > ### Author Response · Authors · 2023-08-13
> > >
> > > You are correct about the denominator; thank you for pointing that out! The metric UnderCovGap captures the average coverage gap among classes that are undercovered and the correct description for how it is computed is:
> > >
> > >
> > > Let $\mathcal{Y}_{\text{under}} = \\{y: \hat{c}_y \leq 1-\alpha \\}$ be the set of classes with coverage less than $1-\alpha$.
> > > Then,
> > >
> > > $$\text{UnderCovGap} = 100 \times \frac{1}{|\mathcal{Y_{\text{under}}}|}\sum_{y \in \mathcal{Y_{\text{under}}}}  |\hat{c}_y -(1-\alpha)|$$

---

> > > > ### Comment · Reviewer_MBi5 · 2023-08-13
> > > >
> > > > Sure, thanks for confirming. Following the discussion period, I've decided to increase my score.

---

### Official Review · Reviewer_omUB · 2023-07-04

**Soundness:** 2 fair
**Presentation:** 3 good
**Contribution:** 2 fair
**Rating:** 4
**Confidence:** 3

**Summary:**

This article studies the clustered conformal prediction, mitigating the issues in the standard conformal and class-wise conformal, by grouping some similar classes together.

**Strengths:**

* The authors proposed clustered conformal prediction to strike the balance between marginal coverage and class-wise coverage for the setting of many classes with limited data per class.
* When calibration data is limited, the empirical results show that the proposed method achieves a relatively small coverage gap according to the designed metric (CovGap), effectively balancing marginal coverage and class-wise coverage.

**Weaknesses:**

* The authors need to specify the low-data scenario (at least on the calibration data from the perspective of the experiments) in the title, otherwise, there are other works related to many classes. Moreover, from the main manuscript, it looks more like cluster-conditional conformal instead of class-conditional conformal in the title.

* The setting and the numerical study somehow contradict each other. Now that the authors used deep learning to train the score function, it does not make sense that there is a limited calibration dataset in practice. If that, why don’t allocate some from the training data?

* There is no sensitivity analysis on the clustering procedure. What if different schemes are used for choosing $M$?  How does $M$ (e.g., $M < K$ or $M > K$) affect the results?

* Using CovGap as a metric might obscure the issue of coverage for each class from methods, e.g., the standard and the proposed method. The results are difficult to say without explicitly showing the coverage for each class. For example, class-wise conformal has high coverage than $1-\alpha$ with the larger gap, but the proposed method could return the coverage below $1-\alpha$ with a smaller gap.


**Questions:**

* The strikingly large prediction set of class-wise conformal is due to the threshold set as in Line 115, but why don’t we set it as the largest score in the calibration data instead of the infinity? There might be a loss of coverage, but it is unclear how large it is in these empirical results, and whether or not this kind of loss is tolerable, especially when the other two competing methods may also lose the coverage for the individual class.

* Line 149: It seems the idea of finding the threshold for the null cluster is the same as that for the marginal coverage. Now that you criticized the standard conformal’s overall coverage ability, why not merge some clusters until you get the desired sample size instead of merging all clusters in the proposed method?

* Line 176: The authors also notice the issue of class imbalance may affect the clustering, then can we up-sample the minor classes to mitigate the issue mentioned in Line 115 and then follow the class-conditional conformal instead of the proposed clustered conformal? Or what if you do the clustering on the original dataset to obtain the desired sample size, and then use class-conditional conformal?


**Limitations:**

Please see the above two parts.

---

> ### Author Rebuttal · Authors · 2023-08-09
>
> Thank you for your review and questions.
>
> In response to weaknesses:
>
> >	W1: “Low-data scenario”;  cluster-conditional conformal vs. class-conditional conformal
>
> We are not aware of other works that focus on creating prediction sets that target class-conditional coverage in the many-class regime. If you are aware of work in this setting, please share! As mentioned in our Related Work section, existing work that evaluates class-conditional coverage focuses on classification problems with at most 10 classes, which is still an order of magnitude off from the setting we work in (100-1000 classes).
>
> Our goal is to produce prediction sets with good class-conditional coverage. We perform clustering as a means towards this goal. The choice of titling the manuscript “cluster-conditional conformal” or “class-conditional conformal” is a matter of emphasizing the mechanism behind the procedure or the result of the procedure. We chose the title of “class-conditional conformal” to highlight this desirable property of the resulting prediction sets.
>
> >	W2: Reallocating training data to increase calibration data
>
> This is a good point. There are many reasons that the amount of calibration data per class may be limited in practice. Nowadays, it is common that models are first pre-trained on large amounts of data that are not from the same distribution that you will deploy the model on and then fine-tuned on a much smaller dataset from the target distribution. Moving data from the pre-training data to the conformal calibration dataset will not lead to valid conformal prediction sets due to the distribution shift. Even if you do have a reasonably large labeled dataset from your target distribution, it is generally undesirable to exclude large amounts of data from training: suppose that we want 100 calibration examples per class but we have 1,000 classes; we would have to set aside 100,000 examples. Removing such large amounts of data from the training dataset will often reduce model accuracy.
>
> We thank the reviewer for raising this point, and we will explicitly raise this in the manuscript.
>
> >	W3: Sensitivity analysis
>
> You can find the sensitivity analysis on the clustering procedure in Appendix B.2. We find that our method is robust to the choice of $M$.
>
> >	W4: CovGap and potential undercoverage
>
> This is another important point, and we have carried out an extensive experiment to investigate this. Please see the common response for details.
>
>
> In response to questions:
>
> >	Q1: Using largest calibration score rather than infinity
>
> As you point out, your suggested modification will not yield a coverage guarantee. To test the effect in practice, we ran an additional experiment following your suggestion. The tl;dr is that (1) as expected, the modified classwise conformal procedure does have smaller set size and results in undercoverage and (2) clustered conformal still does better than modified classwise in terms of both set size (lower AvgSize) and class-conditional coverage (lower CovGap).
> In more detail: (CIFAR-100, with 1000 total calibration points sampled using fixed random seed). FracUnderCov is the fraction of classes with less than 80% coverage.
> * `Classwise`: CovGap = 7.8, AvgSize = 46.1, FracUnderCov = 0.08
> * `Modified Classwise`: CovGap = 7.9, AvgSize = 18.0, FracUnderCov = 0.18
> * `Clustered`: CovGap = 4.5, AvgSize = 8.2, FracUnderCov = 0.08
>
> >	Q2: Null cluster; merging clusters
>
> How we treat classes in the null cluster makes very little practical difference. With that said, one way to get cluster-conditional coverage even for the null cluster is to treat it like any other cluster and estimate a conformal quantile for that cluster. We choose to not do this and opt instead to use the threshold that provides marginal coverage since it is lower variance and works well in practice.
>
> >	Q3: Alternative methods for handling with rare classes
>
> We would not gain anything by up-sampling rare classes and running class-conditional conformal on the synthetically generated sampled dataset. First, from a theoretical perspective, this would not come with any coverage guarantee. The issue is that upsampling the data results in a data set that is not exchangeable with test data – there will be many duplicates in the calibration data, but we know that with probability 1 we will not see any duplicates in the test data. Thus, upsampling would fight against the guarantees underlying conformal prediction.
>
> Second, from a practical perspective, the resulting estimated conformal thresholds would be extremely volatile. Regarding your suggestion to “do the clustering on the original dataset to obtain the desired sample size, and then use class-conditional conformal?” — this is a very reasonable idea and is in fact what we do! What is special about our clustering is that the clusters we produce make it so that we will get good class-conditional coverage and not just cluster-conditional coverage.

---

> > ### Comment · Reviewer_omUB · 2023-08-14
> >
> > Thanks for the author’s explanation.
> >
> > I admit current works like [7, 8, newRef] about class-conditional conformal of many classes only conduct experiments on at most 10 classes. However, they directly control the coverage for each original class without bothering with the extra clustering step. In contrast, like in your discussion section, the necessity of the clustering step is due to the regime of the insufficient calibration data. The gain would be marginal if $n_{avg}$ is larger from the trend of current experiments, or say greater than 200, or even higher (which is not included in the experiments).
> >
> > Theoretically, it directly controls the cluster-wise coverage as in Proposition 1, and may control the class-wise coverage under the requirement/assumption of the clustering as in Proposition 2-3, where total variation $\epsilon$ seems a critical point about the goodness of the clustering step. I feel the authors may need to properly add some discussions on Proposition 3 (although the appendix includes the proof), otherwise, this part is somehow less solid in the main article.
> >
> > By the way, $n_{avg}$ shows up many times but there is no explicit introduction when it first appears, which may confuse the readers.
> >
> > [newRef] Sadinle, Mauricio, Jing Lei, and Larry Wasserman. "Least ambiguous set-valued classifiers with bounded error levels." Journal of the American Statistical Association 114.525 (2019): 223-234.

---

> > > ### Author Response · Authors · 2023-08-15
> > >
> > > We agree with everything you said about when clustering is/isn’t beneficial. If you have a calibration set that is at least moderately large and has only ~10 classes, you should run classwise conformal, as clustering will not provide a benefit. We hoped to have communicated this in our paper, but we can make this more clear in the camera-ready draft. We would also like to note that following reviewer MBi5’s suggestion, we have added discussion of Sadinle et al., 2019 to our related work section, and are happy to add more references too.
> > >
> > > Regarding the issue of low data: our motivation for this work was to achieve good class-conditional coverage on ImageNet (a “many classes” setting) using their validation set of 50,000 images. 50,000 images is not “low data” in aggregate, but divided amongst 1,000 classes, this yields only 50 images per class. Many real applications in computer vision are similar in this regard (reasonably large calibration sets but also "many classes" — hundreds, thousands, or more); in fact, many applications are even worse off than ImageNet in terms of the number of examples available per class. This is where clustering will be useful, since it allows us to dynamically pool information between classes. Following our work, some of our computer vision colleagues are already investigating incorporating clustering into their conformal systems.
> > >
> > > As for the comment about the propositions, thank you, we can explain the role of Propositions 2-3 more carefully in the camera-ready version, including the role of $\epsilon$. We would like to note that, since the initial version, we have been able to further strengthen Proposition 3. Now the same conclusion holds as written, but we only require the KS (Kolmogorov-Smirnov) distance between all pairs of class score distributions within a cluster to be bounded by $\epsilon$, and not the TV (total variation) distance. This is a weaker requirement since the KS distance is never larger than the TV distance.
> > >
> > > Thank you also for the suggestion to explain $n_{\text{avg}}$ more. We have added the following sentence to Line 200 preceding our first use of $n_{\text{avg}}$: “We construct calibration sets of varying size by changing the average number of points in each class, denoted $n_{\text{avg}}$.”
> > >
> > > We hope this helps clarify our reasoning for the positioning of this paper, the choice of experiments, and so on.

---

### Official Review · Reviewer_aaQt · 2023-07-06

**Soundness:** 3 good
**Presentation:** 4 excellent
**Contribution:** 3 good
**Rating:** 7
**Confidence:** 4

**Summary:**

The paper studies how to achieve class-conditional coverage (in the setting of conformal prediction) for multiclass classification problems, in particular for tasks with large label spaces. Previous techniques either provide no class-wise guarantees or tend to produce conservative prediction sets due to lack of data.

The key insight here is that non-conformity scores from different classes may be grouped together if their class-conditional score distributions are similar, since the $(1- \alpha)$-quantile value to achieve desired coverage would be the same across these classes. Essentially, we can extrapolate about a class with low amounts of data by simply using data from other classes with a similar score distribution.

This work is particularly applicable in settings where there is a large number of classes (and thus higher likelihood of small amount of data for some of these classes) and / or high class imbalance.


**Strengths:**

•	Firstly, I found this paper particularly well-written and structured; was written to make the discussion quite intuitive and easy to follow.

•	Method is a natural solution in settings where there is an underlying structure / similarity in how the base model performs on certain groups of classes.

•	Strikes a balance between getting meaningful prediction sets (not overly conservative) and getting low variance in class-wise coverage rates.

•	Empirically, the proposed method performs quite well, performing on par or better than the baseline methods for the studied metrics.


**Weaknesses:**

•	This performance of CLUSTERED seems to be essentially a data and model-driven problem. If there are different classes with similar score distribution, it improves performance.  Is there a possibility of artificially inducing clusters, if such a similarity doesn’t exist?

•	The CovGap metric seems to penalize both overcoverage and undercoverage without distinguishing between the two. It would be nice to see how much each of the methods overcover and undercover (classwise) separately. For example, in Table 2 I would expect that the high CovGap values for CLASSWISE is mostly due to overcoverage, while for STANDARD it is probably undercovering and overcovering in equal measure, but this metric hides that potential distinction.

•	Minor typo: use of $h$ in line 146, equation after line 146 and in Proposition 1 should be $\hat{h}$.


**Questions:**

•	Though equation (2) implies equation (1), in general we may not achieve the guarantee of (2) as you have mentioned in Section 2.2, so the CLUSTERED algorithm may not necessarily achieve marginal coverage. Do you see this method as more of on a continuum with STANDARD and CLASSWISE, with each of them having situations where they work better, especially since CLUSTERED guarantees are not as consistent as the others?

•	Clarification about the usage of Algorithm CLASSWISE: when generating a prediction set for a new input, to choose a conformal predictor to use, we need to know the true label, which we do not. So which of the K predictors is chosen? Do we run all of them and choose the most conservative quantile value?


**Limitations:**

•	The theoretical guarantee on class-conditional coverage depends on the value of $\epsilon$ defined in Proposition 3 which can vary based on the actual score distributions across classes, the value of $m$ used and the clustering algorithm used, so there may be concern on achieving close to $(1 - \alpha)$-coverage performance in general, although in all experiments, the method seems to work quite well.

---

> ### Author Rebuttal · Authors · 2023-08-09
>
> Thank you for your nice summary of the paper!
>
> In response to weaknesses:
>
> >	W1: Inducing artificial clusters
>
> Our method does, in a sense, induce artificial clusters, but we do not view this as a weakness. In the original datasets, there is no true clustering. We simply partition the classes in the way that the data suggests in order to yield good class-conditional coverage.
>
> >	W2: CovGap
>
> We thank the reader for this point. We have conducted a substantial additional experiment to investigate this. Please see the file attached to the common response for plots that separately show undercoverage and overcoverage.
>
>
> In response to questions:
>
> >	Q1: Marginal coverage
>
> Clustered conformal may not always yield perfect class-conditional coverage, but it always yields cluster-conditional coverage (see Proposition 1), which is a stronger guarantee than marginal coverage. Whether cluster-conditional coverage implies class-conditional coverage is dependent on the quality of the clustering (see Proposition 3).
>
> To better clarify this point, we will update the exposition surrounding Proposition 1 and 3 in the text. We thank the reviewer for this point.
>
> >	Q2: Classwise conformal
>
> For a more technical explanation of how the conformal sets are formed by the CLASSWISE procedure, please see Lines 111-112. In words: We first compute a separate conformal quantile for each class $y$. Then, given an input $X$, we compute the conformal prediction set as follows: for each possible label $y$, we compute the conformal score $s(X,y)$ and then include $y$ in the prediction set if $s(X,y)$ is less than the conformal quantile for class $y$.

---

> > ### Comment · Reviewer_aaQt · 2023-08-17
> >
> > Thank you for your detailed response. I have gone through the added experimental results and the other reviewers comments, and would like to maintain my original score.

---

### Official Review · Reviewer_xNf7 · 2023-07-26

**Soundness:** 3 good
**Presentation:** 3 good
**Contribution:** 2 fair
**Rating:** 6
**Confidence:** 3

**Summary:**

The paper describes a method for conformal prediction specially addressed for the case where there is few avaliable data in some of the classes. To address this problem, the proposed method performs a clustering of classes with similar score distributions with the goal of increasing the data used to select the set of classes for a given sample. Experimental results on standard image datasets with a large number of classes with different settings of data availability are reported comparing the proposed method with other standard approaches for conformal prediction.

**Strengths:**

1. The paper addresses the problem of conformal prediction with a large number of classes and with few data per classe, and proposes a new method specifically designed for this setting. The method is well motivated and formalized.
2. Extensive experiments are performed on different datasets and different settings of data availability, comparing the proposed method with standard and classwise approaches. Results show that the proposed method achieves a better average coverage.
3. The paper include useful practical recommendations about when to use each of the conformal prediction approaches.

**Weaknesses:**

1. One of the motivating points of the paper is that, in some cases average coverage can be good enough, but specific coverage for some underepresented classes can be very low. Results show that average coverage improves with the proposed method, but there is not any discussion or results about the specific coverage for particular classes, specially those with few data. Therefore, we cannot validate the original motivation of the paper. Furthermore, as fas as I understand, classes with few data will most probably assigned to the null cluster and then, those classes will follow the standard approach, and there will be no difference between the proposed method and the standard one for these classes.
2. Although the average coverage improves, the AvgSize measure is, in general worse than with the standard method.
3. Some technical details could be better explained or motivated.
- In section 2.1, what do you exactly mean by "scores for all classes in a given cluster are exchangeable? That score distributions are equal?
- The k-means clustering algorithm is very dependent on the selected number of clusters. Although there is a explanation on how the number of cluster is defined, wouldn't it be better to use an adaptive clustering algorithm where the number of clusters can be automatically determined?
- The representation used to perform the clustering is based on a few discrete number of quantile scores. Wouldn't it be better to use a larger and more dense number of quantile scores that could give a better approximation of the distribution of score values?

3. Related work is very oriented to describe on which types of data conformal prediction has been applied. I miss a more technical discussion describing the different techniques and methods used in the existing works.

**Questions:**

See above, specially questions regarding performance on specific classes with few data.

-----------------------------

I have read the rebuttal and it has clarified most of the questions I had about the paper, specially those related with classes with few samples and technical issues. After reading all comments and discussion, I increase my rating to weak accept.

**Limitations:**

The authors discuss on the limitations of the method and no potential negative impacts are foreseen

---

> ### Author Rebuttal · Authors · 2023-08-09
>
> Thank you for your review and, in particular, for highlighting our practical recommendations, which we hope will provide useful guidance for practitioners.
>
> In response to weaknesses:
>
> >	W1: Classes with few examples
>
> See our common response, which describes additional experiments we ran to look at potential undercoverage. It is true that classes with very few examples (say, 5) will end up in the null cluster. In our practical recommendation in the Discussion, we highlight that in regimes where we expect many classes to have very few examples, one should run standard conformal. However, if most classes have few, but not very few, examples (say, 20), clustered conformal will provide a boost in performance.
>
> >	W2: AvgSize of clustered conformal is larger than standard conformal
>
> This is true — there is no free lunch! To get class-conditional coverage, clustered has to output sets that are slightly larger than standard conformal. Standard conformal is not guaranteed to yield class-conditional coverage and can have bad class-conditional coverage in some settings. In many cases, the trade-off of slightly larger sets in return for class-conditional coverage is worth it. Compared to the classwise conformal procedure, we get similar class-conditional coverage with much smaller sets.
>
> >	W3: Adaptive clustering algorithm
>
> We agree that an adaptive clustering algorithm seems nice in theory. However, implementing this in our setting is trickier than it might initially appear. There are several methods designed for choosing $k$ in k-means (e.g., elbow, gap statistic), and we experimented with many of them. However, none of them worked well in our particular clustering setting. The problem of clustering distributions is relatively unexplored and would be interesting to dive into, but it was not the focus of our work. Combining this with the fact that our method is robust to the choice of $k$ in the k-means procedure (see our sensitivity analysis in Appendix B.3) made it reasonable to just use the intuitive procedure for choosing $k$ that we described in the paper.
>
> >	W4: Representation used for clustering
>
> Yes, there is flexibility in choosing the representation upon which to perform clustering. We chose one that makes intuitive sense and stuck with it, since we found it to work well in practice. Increasing the number of quantiles used to create the representation is not necessarily better: for example, if there are only 10 examples, the 85% quantile and the 90% quantile will correspond to the same example.
>
> >	W5: Related work
>
> Thanks for this suggestion; we have expanded our related work. We originally focused on datasets in previous work since the only existing techniques/methods are STANDARD and CLASSWISE, which we described earlier on in the Introduction. CLASSWISE is a special case of Mondrian CP (Line 85), which we now present in more detail. For some related problems (such as creating prediction sets with group-conditional coverage), we have also added some additional discussion of techniques used in previous works.

---

> > ### Comment · Reviewer_xNf7 · 2023-08-18
> >
> > Dear authors,
> > Thank you for your response. It has clarified most of the questions I had about the paper, specially those related with classes with few samples and technical issues. After reading all comments and discussion, I will increase my rating to weak accept.

---

### Author Rebuttal · Authors · 2023-08-09

We would like to thank the reviewers for engaging with our paper and for providing helpful comments and suggestions. We have incorporated the feedback into the paper, and have run a new set of experiments (described below).

The reviewers agreed the paper was well-written and that we thoroughly validated our method via extensive experiments on several datasets.

A couple points we would like to address to all reviewers:

* Several reviewers expressed concerns that the CovGap metric could obscure undercoverage of some classes. To investigate this, we ran experiments that looked specifically at the fraction of classes with less than 80% coverage (FracUnderCov). These results are included in Appendix C.2 and we find that clustered conformal also does well in this metric and  trends in FracUnderCov generally mirror the trends in CovGap. Additionally, in the attached pdf we have included new plots that show the amount of undercoverage (UnderCovGap) and the amount of overcoverage (OverCovGap). More explicitly, UnderCovGap is CovGap restricted to classes with less than the desired coverage level: $$\text{UnderCovGap} = 100 \times \frac{1}{|\mathcal{Y}|}\sum_{y \in \mathcal{Y}}  |\hat{c}_y -(1-\alpha)| \cdot  \mathbf{1}\\{\hat{c}_y \leq (1-\alpha)\\}$$ where $\hat{c}_y$ is the coverage of class $y$. OverCovGap is computed analogously but with $\mathbf{1}\\{\hat{c}_y \geq (1-\alpha)\\}$.
* The goal of our paper is not to champion the use of the clustered conformal method in all settings (and we hope our writing reflects this, particularly in the discussion section). We perform extensive experiments to understand when baselines do well, and we identify a regime in which we can be smarter about how we use our data — in this regime, we recommend the use of clustered conformal.

We provide individual replies to specific comments from each reviewer.

---

### Decision · Program_Chairs · 2023-09-21

**Decision:**

Accept (poster)

**Comment:**

In this paper, the authors investigate conformal prediction that ensures class-conditional coverage guarantees for each class in a multi-class classification problem with numerous classes. It is widely recognized as a significant limitation that conventional conformal prediction can only provide marginal coverage guarantees. This paper can be regarded as one of the studies that overcome this limitation. The authors' main idea is to improve class conditional coverage, particularly for small-sized classes, by clustering similar classes together. All reviewers find the main idea of the proposed method interesting. During the rebuttal and discussion period, most of the reviewers' concerns have been addressed.